# DAG Matters! GFlowNets Enhanced Explainer for Graph Neural Networks

**Wenqian Li**[1][*]**, Yinchuan Li**[2][*]**, Zhigang Li**[3]**, Jianye Hao**[2,3]**, Yan Pang**[1][†]
[1]National University of Singapore, Singapore
[2]Huawei Noah's Ark Lab, Beijing, China
[3]Tianjin University, Tianjin, China
`Wenqian@u.nus.edu,{liyinchuan,haojianye}@huawei.com`
`scs_lzg@tju.edu.cn, jamespang@nus.edu.sg`

## Abstract

Uncovering rationales behind predictions of graph neural networks (GNNs) has received increasing attention over the years. Existing literature mainly focus on selecting a subgraph, through combinatorial optimization, to provide faithful explanations. However, the exponential size of candidate subgraphs limits the applicability of state-of-the-art methods to large-scale GNNs. We enhance on this through a different approach: by proposing a generative structure – GFlowNets-based GNN Explainer (GFlowExplainer), we turn the optimization problem into a step-by-step generative problem. Our GFlowExplainer aims to learn a policy that generates a distribution of subgraphs for which the probability of a subgraph is proportional to its' reward. The proposed approach eliminates the influence of node sequence and thus does not need any pre-training strategies. We also propose a new cut vertex matrix to efficiently explore parent states for GFlowNets structure, thus making our approach applicable in a large-scale setting. We conduct extensive experiments on both synthetic and real datasets, and both qualitative and quantitative results show the superiority of our GFlowExplainer.

## 1 Introduction

Graph Neural Networks (GNNs) have received widespread attention due to the springing up of graph-structured data in real-world applications, such as social networks and chemical molecules Zhang et al. (2020). Various graph related task are widely studied including node classification Henaff et al. (2015); Liu et al. (2020) and graph classification Zhang et al. (2018). However, uncovering rationales behind predictions of graph neural networks (GNNs) is relatively less explored. Recently, some explanation approaches for GNNs have gradually stepped into the public eye. There are two major branches of them: instance-level explanations and model-level explanations Yuan et al. (2022). In this paper, we mainly focus on instance-level explanations.

Instance-level approaches explain models by identifying the most critical input features for their predictions. They have four sub-branches: Gradients/Features-based Zhou et al. (2016); Baldassarre & Azizpour (2019); Pope et al. (2019), Perturbation-based Ying et al. (2019); Luo et al. (2020); Schlichtkrull et al. (2020); Wang et al. (2020), Decompose-based Baldassarre & Azizpour (2019); Schnake et al. (2020); Feng et al. (2021) and Surrogate-based Vu & Thai (2020); Huang et al. (2022); Yuan et al. (2022). Some works such as XGNN Yuan et al. (2020) and RGExplainer Shan et al. (2021) apply reinforcement learning (RL) to model-level and instance-level explanations. However, the pioneering works have some drawbacks. Perturbation-based approaches return the discrete edges for explanations, which are not as intuitive as graph generation-based approach, which could provide connected graphs. However, the task of searching connected subgraphs is a combinatorial problem, and the potential candidates increase exponentially, making most current approaches inefficient and

---

[*]Equal Contribution
[†]Corresponding Author: Yan Pang
This is the joint work between National University of Singapore and Huawei Noah's Ark Lab.
This work was completed while Wenqian Li was a member of the Huawei Noah's Ark Lab for advanced study.

intractable in large-scale settings. In addition, current research consider Monte-Carlo tree search, which has high variance and ignores the fact that graph is an unordered set. This could lead to a loss of sampling efficiency and effectiveness, i.e., the approaches fail to consolidate information of sampled trajectories that form the same subgraph with different sequences.

To address the above issues, we take advantage of the strong generation property of *Generative Flow Networks* (GFlowNets) Bengio et al. (2021b) and cast the combinatorial optimization problem as a generation problem. Unlike the previous work, which focus on the maximization of mutual information, our insight is to learn a generative policy that generates a distribution of connected subgraphs with probabilities proportional to their mutual information. We called this approach `GFlowExplainer`, which could overcome the current predicament for the following reasons. First, it has a stronger exploration ability due to its flow matching condition, helping us to avoid the trap of suboptimal solutions. Second, in contrast to previous tree search or node sequence modeling, GFlowExplainer consolidate information from sampled trajectories generating the same subgraph with different sequences. This critical difference could largely increase the utilization of generated samples, and hence improve the performance. Moreover, by introducing a cut vertex matrix, GFlow-Explainer could be applied in large-scale settings and achieve better performance with fewer training epochs. We summarize the main contributions as follows.

**Main Contributions:** 1) We propose a new hand-crafted method for GNN explanation via GFlowNet frameworks to sample from a target distribution with the energy proportional to the predefined score function; 2) We take advantage of the DAG structure in GFlowNets to connect the trajectories of outputting the same graph but different node sequences. Therefore, without any pre-training strategies, we can significantly improve the effectiveness of our GNN explanations; 3) Considering relatively cumbersome valid parent state explorations in GFlowNets because of the connectivity constraint of the graph, we introduce the concept of cut vertex and propose a more efficient cut vertex criteria for dynamic graphs, thus speeding up the whole process; 4) We conduct extensive experiments to show that GFlowExplainer can outperform current state-of-the-art approaches.

## 2 RELATED WORK

**Graph Neural Networks:** Graph neural networks (GNNs) are developing rapidly in recent years and have been adopted to leverage the structure and properties of graphs Scarselli et al. (2008); Sanchez-Lengeling et al. (2021). Most GNN variants can be summarized with the message passing scheme, which is composed of pattern extraction and interaction modeling within each layer Gilmer et al. (2017). These approaches aggregate the information from neighbors with different functions, such as mean/max/LSTM-pooling in GCN Welling & Kipf (2016), GrpahSAGE Hamilton et al. (2017), sum-pooling in GIN Xu et al. (2018), attention mechanisms in GAT Velickovic et al. (2017). SGC Wu et al. (2019) observes that the superior performance of GNNs is mainly due to the neighbor aggregation rather than feature transformation and nonlinearity, and proposed a simple and fast GNN model. APPNP Klicpera et al. (2018) shares the similar idea by decoupling feature transformation and neighbor aggregation.

**Generative Flow Networks:** Generative flow networks Bengio et al. (2021a;b) aim to train generative policies that could sample compositional objects $x \in \mathbb{D}$ by discrete action sequences with probability proportional to a given reward function. This network could sample trajectories according to a distribution proportional to the rewards, and this feature becomes particularly important when exploration is important. The approach also differs from RL, which aims to maximize the expected return and only generates a single sequence of actions with the highest reward. GFlowNets has been applied in molecule generation Bengio et al. (2021a); Jain et al. (2022), discrete probabilistic modeling Zhang et al. (2022), bayesian structure learning Deleu et al. (2022), causal discovery Li et al. (2022) and continuous control tasks Li et al. (2023).

**Instance-level GNN Explanation:** Instance-level approaches explain models by identifying the most critical input features for their predictions. Gradients/Features-based approaches, e.g., Zhou et al. (2016); Baldassarre & Azizpour (2019); Pope et al. (2019), compute the gradients or map the features to the input to explain the important terms while the scores sometimes could not reflect the contributions intuitively. As for the perturbation-based approaches, GNNExplainer Ying et al. (2019) is the first specific design for explanation of GNNs, which formulates an optimization task to maximize the mutual information between the GNN predictions and the distribution of poten-

tial subgraphs. Unfortunately, GNNExplainer and Causal Screening Wang et al. (2020) may lack a global view of explanations and be stuck at local optima. Even though PGExplainer Luo et al. (2020) and GraphMask Schlichtkrull et al. (2020) could provide some global insights, they require a reparameterization trick and could not guarantee that the outputs of the subgraph are connected, which lacks explanations for the message passing scheme in GNNs. Shapley-value based approaches SubgraphX Yuan et al. (2021) and GraphSVX Duval & Malliaros (2021) are computationally expensive especially for exploring different subgraphs with the MCTS algorithm. Decomposed-based approaches, for example, LRP Baldassarre & Azizpour (2019), GNN-LRP Schnake et al. (2020) and DEGREE Feng et al. (2021), evaluate the importance of input features by decomposing the model predictions into several terms , at the price of raising the difficulty of applying the method to complex and structured graph datasets. Surrogate-based approaches PGMExplainer Vu & Thai (2020) and GraphLime Huang et al. (2022) sample a data set from the neighbors of a given example and then fit an interpretable model for that data set. However, this approach requires a careful definition of neighboring areas, making the generalization to other problem settings highly non-trivial. As an another attempt, XGNN Yuan et al. (2020) and RGExplainer Shan et al. (2021) apply reinforcement learning to model-level and instance-level explanations respectively, while the latter requires inefficient pre-training strategies and has high variances for sampling.

# 3 GFLOWEXPLAINER

## 3.1 PROBLEM FORMULATION

Let $G = (\mathcal{V}, \mathcal{E})$ denote a graph on nodes $\mathcal{V}$ and edges $\mathcal{E}$ with $d$-dimensional node features $\mathcal{X} = \{x_1, ..., x_n\}, x_i \in \mathbb{R}^d$. The adjacency matrix $A$ describes the edge relationships of $G$, i.e., $A_{ii} = 1$ for all $i \in \mathcal{V}$ and $A_{ij} = 1$ for all $\{v_i, v_j\} \in \mathcal{E}$. $\hat{A}$ is the symmetrical adjacency matrix computed by $\hat{A} = \tilde{D}^{-1/2} A \tilde{D}^{-1/2}$ where $\tilde{D}$ is the diagonal degree matrix of $A$. Let $\Phi$ denote a trained GNN model, which is optimized on all instances in the training set and is then used for predictions. Given an instance, i.e. a node $v$ or a graph $G$, the goal of GNN Explanation is to identify a subgraph $G_s = (\mathcal{V}_s, \mathcal{E}_s)$ and the associated features $X_s = \{x_j | v_j \in G_s\}$ that are important for the GNN prediction $Y_i = \Phi(v_i)$ or $Y_{g_i} = \Phi(G_i)$ where $g_i$ is a graph instance. The previous works formulate this task as an optimization problem and the objective is to maximize the mutual information

$$\max_{G_s} MI(Y, G_s) = H(Y) - H(Y|G_s) \iff \min_{G_s} H(Y|G_s), \quad (1)$$

where $MI(\cdot)$ is the mutual information function, $H(\cdot)$ is the entropy function, $\hat{y}$ is the prediction of $\Phi$ with $G_s$ as the input and $H(Y|G_s) = -\mathbb{E}_{Y|G_s}[\log P_\Phi(Y|G_s)]$. Since $H(Y)$ is fixed in the explanation state, the objective can be rewritten as $\min_{G_s} H(Y|G_s)$, which is to minimize the uncertainty of $\Phi$ when the GNN computation is limited to $G_s$.

From the graph generation perspective, since there are exponential candidates for explaining for $\hat{y}$, it is not trivial to direct solve such combinatorial optimization problem. Thus we turn this optimization problem into a step-by-step generative problem (see Figure 1). We propose our generative structure as GFlowNets-based GNN Explainer, abbreviated as `GFlowExplainer`, which consists of a tuple $(\mathcal{S}, \mathcal{A})$ where $\mathcal{S}$ is a finite set of states, and $\mathcal{A}$ is the action set consisting transitions $a_t : s_t \rightarrow s_{t+1}$. The insight comes from that we could consider $G_s$ as a compositional object. Starting from an empty graph, we can train our policy network to generate such $G_s$ by sequentially adding one neighbor node at each step $t$ to ensure the connectivity of an explanation graph, in which $G_s(s_t)$ refers to a subgraph at state $s_t$, and adding one node refers to an action $a_t$ making a state transition $s_t \rightarrow s_{t+1}$. Different from traditional optimization problems maxmizing the mutual information our objective is to construct a TD-like flow matching condition, to obtain a generative forward policy $\pi(a_t|s_t)$ so that $\mathcal{P}(Y, G_s) \propto r(Y, G_s)$, where $r(Y, G_s)$ is a predefined reward function based on $MI(Y, G_s)$.

The rest of the section is organised as follows: we first introduce the flow modeling of GFlowNets in Section 3.2. The crucial elements of GFlowExplainer structure are defined in Sections 3.3 and 3.4. We propose a new framework to address the connectivity problem for an effective exploration of parent states in GFlowExplainer in 3.5.

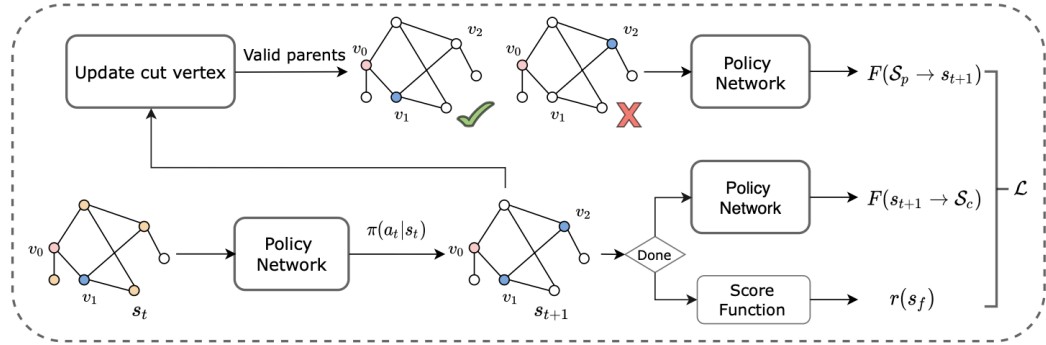

Figure 1: **Structure of GFlowExplainer:** Sampling from a starting node $v_0$ (pink), for each state $s_t$, the combined features of subgraph (pink and black) and neighbor nodes (orange) are fed into the policy network to sample an allowed action $a_t : \{v_2\}^+ \sim \pi(a \mid s_t)$ and obtain $s_{t+1}$. Then cut vertices are updated based on $s_{t+1}$ to find valid parents set $\mathcal{S}_p$ for calculating inflows $F(\mathcal{S}_p \rightarrow s_{t+1})$. Either outflows $F(s_{t+1} \rightarrow \mathcal{S}_c)$ or reward $r(s_f)$ is calculated based on the stopping criteria.

## 3.2 FLOW MODELING

**Flows and Probability Measures:** Following Bengio et al. (2021b), Consider a direct acyclic graph (DAG) $\mathcal{G} = (\mathcal{S}, \mathcal{A})$, where $\mathcal{S}$ is a finite set of states and $\mathcal{A}$ is a subset of $\mathcal{S} \times \mathcal{S}$ representing directed edges, and each element of $\mathcal{A}$ corresponds to the state transition $a_t : s_t \rightarrow s_{t+1}$. The complete trajectory is a sequence of states $\tau = (s_0, ..., s_n, s_f) \in \mathcal{T}$ where $s_0$ is an initial state, $s_f$ is a terminal state and $\mathcal{T}$ is a set containing all complete trajectories. In order to measure the probabilities associated with states $s$, a non-negative function $F(\cdot)$ corresponding "flow" is introduced. $F(s_t, a_t) = F(s_t \rightarrow s_{t+1})$ corresponds to an edge flow or action flow. $F(\tau)$ corresponds to the trajectory flow and the state flow is the sum of all trajectory flows passing though that state, denoted as $F(s) = \sum_{s \in \tau} F(\tau)$. If we fix the total flow of the DAG as $Z$ flowing into terminal states $s_f$ to the given value $r(s_f)$, and consider the DAG as a water pipe, in which water enters in $s_0$ and flows out through all $s_f$, we can obtain $Z = F(s_0) = \sum F(s_f) = \sum r(s_f)$.

Based on this flow network, the stochastic policy $\pi$ associated with the normalized flow probability $\mathcal{P}$ is defined as follows,

$$\pi(a_t \mid s_t) = \mathcal{P}_F(s_{t+1} \mid s_t) = \mathcal{P}(s_t \rightarrow s_{t+1} \mid s_t) = \frac{F(s_t \rightarrow s_{t+1})}{F(s_t)}, \tag{2}$$

where $\mathcal{P}_F(s_{t+1} \mid s_t)$ is called the forward transition probability. Then we can obtain $\mathcal{P}_F(\tau) = \prod_{t=0}^{t=f-1} \mathcal{P}_F(s_{t+1} \mid s_t)$, which yields

$$\mathcal{P}_F(s) = \sum_{\tau:s \in \tau} \mathcal{P}_F(\tau) = \frac{\sum_{\tau \in \mathcal{T}} \mathbb{I}_{s \in \tau} F(\tau)}{\sum_{\tau \in \mathcal{T}} F(\tau)} = \frac{F(s)}{Z}. \tag{3}$$

Our goal is to obtain $\mathcal{P}_F(s_f) = \sum_{\tau:s_f \in \tau} \mathcal{P}_F(\tau) \propto r(s_f)$.

**State-Conditional Flow Network** Following Bengio et al. (2021b), consider a flow network based on a DAG $\mathcal{G} = (\mathcal{S}, \mathcal{A})$ and a non-negative flow function $F(\cdot)$. For each state $s \in \mathcal{S}$, the subgraph $\mathcal{G}_s$ consists of all $s'$ such that $s' \geq s$, where $\geq$ follows the partial order. Then the state-conditional flow network is based on the family $\{\mathcal{G}_s, s \in \mathcal{S}\}$ with a conditional flow function $F: \mathcal{S} \times \mathcal{T} \rightarrow \mathbb{R}^+$, in which $\mathcal{T} = \cup_{s \in \mathcal{S}} \mathcal{T}_s$ and $\mathcal{T}_s$ is the set of trajectories in $\mathcal{G}_s$ containing all $\{\tau = (s, ..., s_f)\}$ such that $\forall s_n, s_m \in \tau$

$$F_s(s_n \rightarrow s_m) = F(s_n \rightarrow s_m). \tag{4}$$

Based on this definition, we have the initial flow of the state-conditional flow network refers to marginalize the *terminating flows* $F(s' \rightarrow s_f)$, i.e., for any terminating state $s' \geq s$ we have Bengio et al. (2021b)

$$F_s(s_0 \mid s) := F_s(s) = \sum_{s':s' \geq s} F(s' \rightarrow s_f). \tag{5}$$

Then, we can obtain the corresponding probability measures as the following

$$\mathcal{P}_s(s' \mid s) = \frac{F_s(s' \to s_f)}{F_s(s)}, \quad \forall s' \geq s. \tag{6}$$

The flow $F(s)$ through state $s$ in the original flow network could not provide the marginalization over the downstream terminating flows, we thus introduce this state-conditional flow network satisfying the desired marginalization property. In our task, considering the message passing scheme, we set to start sampling the trajectory based on a chosen starting node $v_0$, thus the transition from an empty graph to that node $v_0$ is ignored here. We can consider that each trajectory is sampled from the subgraph family $\{\mathcal{G}_{s_0}\}$, where $s_0 = v_0$. Then similar to the way to estimate the flow of a flow network using GFlowNet, based on the conditional state flow network, we could still train a policy to obtain $\mathcal{P}_{s_0}(s_f) \propto r(s_f)$. Based on equation 4, we can omit this subscript in the following sections.

## 3.3 STATES AND ACTIONS

In this subsection, we give the following definitions on states and actions, and also node neighbours and graph neighbors for following valid action set $\mathcal{A}$ and valid parent states in Section 3.5.

**Definition 1** *(State) A state $s_t \in \mathcal{S}$ in GFlowExplainer refers to a subgragh $G_s(s_t)$ consists of several nodes. The initial state $s_0$ contains a starting point $v_0$ and a final state $s_f$ is a subgraph attaining the stop criteria.*

Since we need to guarantee the connectivity of the generated subgraph $G_s$, for every step we can only select an node from the boundary of the current subgraph $G_s(s_t)$.

**Definition 2** *(Neighbours) There are two types of neighbors: node neighbors and graph neighbors. $\forall v_i, v_j \in G_s$, if $\{v_i, v_j\} \in \mathcal{E}$, then we define $v_i$ as a neighbor of node $v_j$, denoted as $v_i \in \mathcal{N}(v_j)$, and vice versa; $\forall v_i \notin G_s$, if $\exists v_j \in G_s$, such that $\{v_i, v_j\} \in \mathcal{E}$, then we define $v_i$ as a neighbor of graph $G_s(s_t)$, denoted as $v_i \in \mathcal{N}(s_t)$.*

Simply to say, graph neighbours contain boundary nodes that have yet been selected into the subgraph. Node neighbours represent the connect relationships of each pair of nodes in the subgraph.

**Definition 3** *(Action) An action $a_t : s_t \to s_{t+1} \in \mathcal{A}$ in GFlowExplainer is to add a node from $\mathcal{N}(s_t)$, denoted as $a_t : \{v_i\}^+ \sim \mathcal{N}(s_t)$. Thus making a state transition $s_{t+1} = s_t \cup \{v_i\}$.*

Since we need to combine the features of all nodes in $\mathcal{N}(s_t)$ and $G_s(s_t)$ as the input to calculate the action distribution, for each node $v_i \in \mathcal{N}(s_t) \cup G_s(s_t)$, we concatenate two indicator functions with its original feature vector $x_i$, to distinguish the initial node $v_0$ and all nodes in the subgraph $G_s(s_t)$. The insights behind are: 1) for the node classification task, the generated same subgraph should have different scores for the specific node to be explained; 2) the allowed action is to select a node in $\mathcal{N}(s_t)$ instead of $G_s(s_t)$, which will introduce cycles for our DAG structure. Therefore, the initial feature representation $X'_t$ is obtained by follows,

$$x'_i = [x_i, \mathbb{1}_{v_i=v_0}, \mathbb{1}_{\{v_i \in G_s(s_t)\}}], \quad X'_t = [x'_i]_{\forall v_i \in G_s(s_t) \cup \mathcal{N}(s_t)}. \tag{7}$$

Considering the associations among nodes in graph structured data, for each node $v_i$, it is crucial to combine information from its neighbours. To achieve this, we apply APPNP, a GNN method proposed by Klicpera et al. (2018), which separates the non-linear transformation and information propagation. We have the following update equation,

$$H_t^{(0)} = \Theta_1 X'_t, \quad H_t^{(l+1)} = (1-\alpha)\hat{A}H_t^{(l)} + \alpha H_t^{(0)}, \tag{8}$$

where $\Theta_1$ is the trainable weight matrix, $\alpha$ is a hyper-parameter used to control weight. After $L-$layer updates, we obtain the node representations $H_t^L$, and then feed them into a MLP to improve the representation ability:

$$\bar{H}_t(v_i) = \text{MLP}(H_t^L(v_i); \Theta_2), v_i \in G_s(s_t) \cup \mathcal{N}(s_t), \tag{9}$$

where $\Theta_2$ is the learnable parameters in the MLP.

## 3.4 Reward, Starting node and Stop Criteria

Similarly to Luo et al. (2020), we use the cross-entropy function to replace the conditional entropy function $H(Y \mid s_f)$ with $N$ given instances, and define the reward function as follows:

$$r(s_f, Y) = \exp(-\mathcal{L}(s_f, Y)) = \exp(-\frac{1}{N} \sum_{n=1}^{N} \sum_{c=1}^{C} P(Y = c) \log P(\hat{y} = c)), \qquad (10)$$

where $\mathcal{L}$ is the prediction loss; $s_f$ is the generated explanatory subgraph for an instance; $C$ is the number possible predicted labels; $P(\hat{y} = c)$ is the probability that the original prediction of the trained GNN $\Phi$ is $c$; and $P(Y = c)$ is the probability that the label prediction of $\Phi$ on the subgraph $s_f$ is $c$. We use the exponential term here is to avoid negative reward.

For node classification tasks, the starting node is the node instance to be interpreted. In contrast, for graph classification tasks, any node could be the potential starting node and the choice of it determines the explanation performance. Therefore, we construct a locator $\mathbb{L}$ to identify the most influential node in the graph similarly to Shan et al. (2021). Given $N$ graph instances $g_n$, the prediction loss of the classification can be rewritten with the locator $\mathbb{L}$ as $\hat{y} = \Phi(\pi(s_f|\mathbb{L}(g_n)))$. We train a three-layer MLP to model the influence of a node $v_{i,n}$ on the label of the graph instance $g_n$:

$$\omega_{i,n} = \text{MLP}([z_{g_n}, z_{v_{i,n}}]), \qquad (11)$$

where $z_{g_n}$ and $z_{v_{i,n}}$ are respectively the feature representations of the graph $g_n$ and the node $v_{i,n}$ after 3-GCN layers based on the trained model $\Phi(\cdot)$. We train this neural network based on some sampling graph instances with the Kullback-Leibler divergence loss

$$\text{KLDivLoss}(\omega_{i,n}, -\mathcal{L}(\pi(s_f \mid v_{i,n}), Y_{g_n})),$$

so that the distribution between estimated value $\omega_{i,n}$ is closed to $-\mathcal{L}(\pi(s_f \mid v_{i,n}), Y_{g_n})$ asymptotically, and the softmax layers are used to transform these two values into their distributions.

To obtain a compact explanation and avoid generating large subgraphs, we impose a constraint $|s_f| \leq K_M$ so that $s_f$ has at most $K_M$ nodes. We also introduce a self-attention mechanism similarly to Shan et al. (2021), which could aggregate the feature representations:

$$\gamma_t(v_i) = \frac{\exp(\theta_1^T \bar{H}_t(v_i))}{\sum_{v_j \in \mathcal{N}(s_t)} \exp(\theta_1^T \bar{H}_t(v_j))}, v_i \in \mathcal{N}(s_t), \qquad (12)$$

$$\bar{H}_t(STOP) = \sum_{v_i \in G_s(s_t) \cup \mathcal{N}(s_t)} \gamma_t(v_i) \bar{H}_t(v_i), \qquad (13)$$

where the parameter $\theta_1$ learns the attention $\gamma_t(v_i)$ for each node $v_i$. We can concatenate $\bar{H}_t(STOP)$ into feature representations in equation 9. We should note that all learnable parameters above are the components in our policy network.

## 3.5 Efficient Parent State Explorations

Flow matching condition is a crucial element in flow modeling. For current state $s_t$, we need to explore all its direct parent states and corresponding one-step actions, i.e. $s, a : T(s, a) = s_t$, which refers all sets $(s, a)$ that could attain $s_t$. However, the connectivity constraint makes exploring valid parents non-trivial since we need to guarantee that the graph is always connected. We consider this task a cut vertex exploration problem, which aims to find all vertices that will break the connectivity of a graph for each $s_t$. If a node is a cut vertex, we can not find a valid parent state by deleting it. By taking advantage of the step-by-step generative process, we can update and store the cut vertex without repeatedly checking. Based on Definitions 4 and 5, in the following Theorem 1 we show how to update cut vertices for each step, which is proved in Appendix A.2.

**Definition 4** *(Cut vertex matrix) A cut vertex matrix of a state $s_t$ is a dynamic matrix $\mathcal{Z} \in \mathbb{R}^{t \times t}$, where $\mathcal{Z}_{i,i} = 0$ and if $\exists \mathcal{Z}_{i,j}(s_t) \neq 0$, then we say $v_i$ is a cut vertex at $s_t$.*

**Definition 5** *(Connectivity vector) Suppose an action $a_t = \{v_j\}^+$, a connectivity vector of state $s_t$ is a binary vector $z \in \{0, 1\}^{t \times 1}$, $\forall v_i \in G_s(s_t)$, $z_i(s_t) = 1$ if $\{v_i, v_j\} \in \mathcal{E}$.*

**Lemma 1** *Suppose an action $a_t = \{v_j\}^+, v_i \in G_s(s_t)$. If $v_i$ is not a cut vertex at $s_t$, $v_i$ becomes a cut vertex from $s_{t+1}$ iff $|\mathcal{N}(a_t)| = 1$ and $\{v_i, v_j\} \in \mathcal{E}$, where $t > 1$. If $v_i$ is a cut vertex at $s_t$, $v_i$ is not a cut vertex from $s_{t+1}$ iff $|\mathcal{N}(a_t)| > 1$ and $a_t$ connects to all "child groups" of the $v_i$.*

**Theorem 1** *Staring from $\mathcal{Z}(s_t) = [\mathbf{0}]_{2 \times 2}$, for any action $a_t = \{v_j\}^+ \in \mathcal{A}, t \geq 2$, the connectivity vector of state $s_t$ is constructed by*

$$z_k(a_t) = A_{j,k}, \forall v_k \in G_s(s_t), \tag{14}$$

*then we update cut vertex matrix $\mathcal{Z}(s_{t+1})$ by*

$$\mathcal{Z}(s_{t+1}) = \left[ \begin{array}{cc} \mathcal{Z}'(s_t) & z'(a_t) \\ 0 & 0 \end{array} \right], \tag{15}$$

*where $\mathcal{Z}'(s_t)$, $z'(s_t)$ are constructed based on the following equations:*
*1. If $|\mathcal{N}(a_t)| = 1, v_k \in G_s(s_t), \{v_k, v_j\} \in \mathcal{E} : \forall v_m \in G_s(s_t)$*

$$\begin{cases} \mathcal{Z}'(s_t) = (1 - I_{t \times t}) \wedge \{\mathcal{Z}(s_t) + \mathbb{I}_{\mathcal{Z}_k(s_t)[1-z(a_t)]=0} \, z(a_t) \cdot [\mathbf{I}]_{1 \times t}\} \\ z'(a_t) = \mathcal{Z}'(s_t) \cdot z(a_t) + z(a_t) \wedge [\max\{\mathcal{Z}'_m(s_t)\} + 1]_{t \times 1} \end{cases} \tag{16}$$

*2. If $|\mathcal{N}(a_t)| > 1, k = 0, ..., t : \forall v_m \in G_s(s_t)$*

$$\begin{cases} \mathcal{Z}'_{m,k}(s_t) = \mathbb{I}_{set} \left[ \mathbb{I}_{set2} \max\{\mathcal{Z}_m(s_t) \wedge z^T(a_t)\} + (1 - \mathbb{I}_{set2})\mathcal{Z}_{m,k}(s_t) \right] \\ z'_m(a_t) = \mathbb{I}_{sum} \left[ \max\{\mathcal{Z}'_m(s_t) \wedge z^T(a_t)\} \right] \end{cases} \tag{17}$$

*where $\mathbb{I}_{set} = 1$ iif $set(\mathcal{Z}_m(s_t) \wedge z^T(s_t)) \neq set(\mathcal{Z}_m(s_t))$, $\mathbb{I}_{sum} = 1$ iff $\sum \mathcal{Z}'_m(s_t)z(a_t) \neq 0$. $\mathbb{I}_{set2} = 1$ iff $\mathcal{Z}_{m,k}(s_t) \in set(\mathcal{Z}_m(s_t) \wedge z^T(a_t))$. $set(\cdot)$ corresponds to distinct value (except 0) in a vector. Then based on Lemma 1, we have the following criteria to ensure the valid parent exploration in GFlowNets: for $t \geq 2$, $v_i$ is a cut vertex at $s_t$ iff $\exists \mathcal{Z}_{i,j}(s_t) \neq 0$.*

Theorem 1 shows how we utilize dynamic graphs to efficiently update the cut vertex and thus guarantee valid parent explorations. This approach is a kind of "amortized" checking since we only need to consider additional edges from $a_t$ instead of all edges and nodes in the $G_s(s_t)$, thus having lower complexity than previous approaches. We will show the theoretical analysis in Appendix D.3.

## 3.6 TRAINING PROCEDURE

Starting from the starting node, GFlowExplainer draws complete trajectories $\tau = (s_0, s_1, ..., s_f) \in \mathcal{T}$ by iteratively sampling $\{v_i\}^+ \sim \pi(a_t \mid s_t)$, until the stopping criteria is attained. After sampling a buffer, to train the policy $\pi(s_t \mid a_t)$ which satisfies $\mathcal{P}(s_f, Y) \propto r(s_f, Y)$, we minimize the loss over the flow matching condition as follows

$$\mathcal{L}(\tau) = \sum_{s_{t+1} \in \tau} \left( \sum_{T(s_t, a_t) = s_{t+1}} F(s_t, a_t) - \mathbb{I}_{s_{t+1} = s_f} r(s_f, Y) - \mathbb{I}_{s_{t+1} \neq s_f} \sum_{a_{t+1} \in \mathcal{A}} F(s_{t+1}, a_{t+1}) \right)^2, \tag{18}$$

where $\sum_{T(s_t, a_t) = s_{t+1}} F(s_t, a_t)$ denotes the inflows of a state $s_{t+1}$, $\sum_{a_{t+1} \in \mathcal{A}} F(s_{t+1}, a_{t+1})$ denotes the outflows of $s_{t+1}$, and $r(s_f, Y)$ denotes the reward of the final state, which is computed by equation 10. For interior states, we only calculate outflows based on action distributions. For final states, there are no outgoing flows and we only calculate their rewards. We summarize algorithms for both node classification task and graph classification task in Appendix C.

## 4 EXPERIMENTS

In this section, we first introduce our experimental setup. Then we compare GFlowExplainer with a few state-of-the-art baselines GNNExplainer Ying et al. (2019), PGExplainer Luo et al. (2020), DEGREE Feng et al. (2021) and RG-Explainer Shan et al. (2021) in both qualitative and quantitative evaluations. Further, we evaluate the performance of our approach in the inductive setting as well as ablation experiments in Section 4.4 and Appendix D.

## 4.1 Experimental Setup

**Datasets** We use six datasets, in which four synthetic datasets (BA-shapes,BA-Community,Tree-Cycles and Tree-Grid) are used for the node classification task and two datasets (BA-2motifs and Mutagenicity) are used for the graph generation task. These datasets are composed of *motifs* and *bases*. Motifs are small substructures in a graph, which have been shown to play a crucial role in predicting the label of node/graph instances. Bases are the remaining parts of a graph which are randomly generated. Motifs are taken as the ground-truth and the goal of explainers is to find them. Details of these datasets are described in Appendix E.3 and the visualizations are shown in Figure 9.

**Model** We use the trained GNN model in Holdijk et al. (2021), whose architecture is given in Luo et al. (2020); Ying et al. (2019). Specially, for node classification, we use the model which consists of three consecutive graph convolution layers connected with a fully connected layer. For graph classification, the model includes three consecutive graph convolution layers fed into two max and mean pooling layers. The two pooling layer output embeddings are then concatenated to generate the input for a fully connected layer.

**Metrics** The motifs in each dataset are the ground-truth explanations. The edges in the motif are positive and other edges are negative. GNNExplainer and PGExplainer return a mask matrix to represent the importance of each edge in the instance. RGExplainer and ours generate a subgraph. The explanation problem can be formalized as a binary classification task, where edges in the ground-truth motif are taken as prediction labels and the weights of edges are viewed as prediction scores. With the explanatory subgraph provided by explainers, the **AUC score** can be computed to measure the accuracy for quantitative evaluation.

## 4.2 Qualitative Analysis

We evaluate the single-instance explanations for the topology-based prediction task without node features in Figure 2, in which the dots in green are our predicted nodes in motif, representing the critical nodes for GNN predictions. In contrast, the dots in orange are predicted nodes not in motif, referring to the irrelevant nodes for GNN predictions. The pink dot is the node to be interpreted, also included in the subgraph. For a fair comparison, we choose the same node for each algorithm and output their generated subgraphs. As illustrated in the figure, house, cycle, and tree motifs are identified by GFlowExplainer and have relatively fewer irrelevant nodes and edges. However, in the BA-Community dataset, RGExplainer fails to find the motif. For graph classifications, we visualize the explanation result for the BA-2motif dataset, and both approaches could find the five-node cycle motif for label 1.

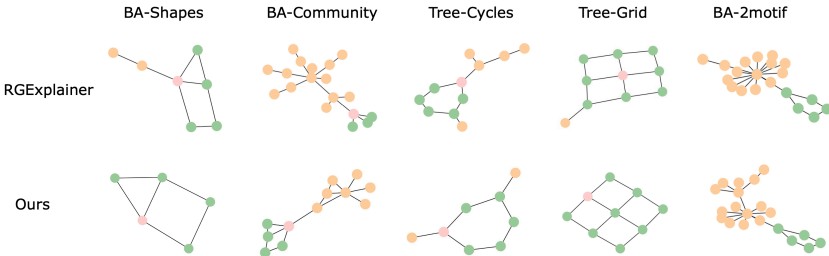

Figure 2: Qualitative Analysis for RGExplainer and GFlowExplainer

## 4.3 Quantitative Analysis

We next show the quantitative results in Table 1. We run 10 different seeds for each approach and compute the average AUC scores and their standard deviations. From the table, we can find that our GFlowExplainer performs the best on five datasets. The difference between GFlowExplainer and the runner-up algorithm is not particularly noticeable on the BA-Shapes and MUTAG datasets. However, on the Tree-Cycles and BA-2motif datasets, GFlowExplainer improves the performance and shows its superiority to other graph generation and perturbation approaches. We should notice that without pre-training, the AUC scores of the RGExplainer on Tree-Cycles and Tree-Grid are always 0.5, while GFlowExplainer does not need any pre-training process and could access the

Table 1: Explanation AUC (Quantitative Evaluation)

| | Node Classification | | | | Graph Classification | |
|---|---|---|---|---|---|---|
| | BA-Shapes | BA-Community | Tree-Cycles | Tree-Grid | BA-2motifs | MUTAG |
| GNNExp | 0.742±0.006 | 0.708±0.004 | 0.540±0.017 | 0.714±0.002 | 0.499±0.001 | 0.498±0.003 |
| PGExp | 0.974±0.005 | 0.884±0.020 | 0.574±0.021 | 0.673±0.004 | 0.133±0.045 | 0.843±0.084 |
| DEGREE | 0.993±0.005 | **0.957±0.010** | 0.902±0.022 | 0.925±0.040 | 0.755± 0.135 | 0.773±0.029 |
| RGExp (NoPretrain) | 0.983±0.021 | 0.684±0.012 | 0.500±0.000 | 0.500±0.000 | 0.503±0.011 | 0.623±0.021 |
| RGExp | 0.985±0.013 | 0.858±0.021 | 0.787±0.099 | 0.927±0.030 | 0.615±0.029 | 0.832±0.046 |
| Ours | **0.999±0.000** | 0.938±0.019 | **0.964±0.028** | **0.982±0.011** | **0.854±0.016** | **0.882±0.024** |
| Improve | 1.4% | -2.0% | 6.8% | 5.9% | 13.1% | 4.6% |

Figure 3: Comparison among GFlow-Squence, GFlowExplainer, RG-NoPretrain and RGExplainer in the inductive setting for synthetic datasets. GFlowExplainer has better generalizations.

ground-truth motif better. Even though on BA-Community datasets, GFlowExplainer is a runner-up algorithm, it is not far away from DEGREE.

## 4.4 INDUCTIVE SETTING WITH ABLATION EXPERIMENTS

To further show the effectiveness of our proposed theorem and the generalization ability of GFlow-Explainer, we conduct the ablation experiments with various cases and test the performance of GFlowExplainer in the inductive setting. We compare GFlowExplainer with GFlow-Sequence, a GFlowNets-based approach with the same state encoding, action space, reward function, and objective function. The difference lies in that the state is considered as a sequence and for each state $s_t$, there is only one parent state $s_{t-1} = s_t/\{v_i\}$, where $a_{t-1} = \{v_i\}^+$, which is similar to RGExplainer. We also compare our GFlowExplainer with RGExplainer-Nopretrain and RGExplainer.

Specifically, we vary the training set sized from $\{10\%, 30\%, 50\%, 70\%, 90\%\}$ and take the remaining instances for testing. For each dataset, we run the experiments 5 times and compute the average AUC scores. For fairness, we set the same parameters for each method. The comparison results are shown in Figure 3. As for BA-Shapes and Tree-Cycles, since they already have enough training samples for GFlowexplainer when the ratio is 10%, the performances of it are always good enough and fall in certain intervals. We also note that in some seeds, RGExplainer dropped sharply from the initial AUC of 0.77 to 0.5. We conjecture that the policy gradient and Monte Carlo estimation may suffer from high variances and be unstable, which may not be able to generate an explanation consistently. In contrast, GFlowExplainer does not need any pre-training strategies and could provide more consistent explanations. Finally, we discuss more the properties of DAG and why it becomes the critical ingredient of best performance in our work in Appendix B.1.

## 5 CONCLUSION

In this work, we present GFlowExplainer to provide the instance-level explanations for GNNs. The DAG structure in our method eliminates the influence of node sequence and thus without any pre-training strategies, we could provide faithful and consistent explanations with the ensurance of the message passing nature of GNNs. We also propose a specific approach for checking cut vertices in dynamic graphs, thus accelerating the process of direct parents exploration during the training process. Extensive experiments confirm the efficiency and strong generative ability of GFlowExplainer.

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

## A    PROOF OF MAIN RESULTS

### A.1    PROOF OF LEMMA 1

**Definition 6** *A node $v_i$ is a cut vertex iif $\exists v_a, v_b \in G$, such that there is no node sequence $\vec{v} = (v_a, ..., v_b)$ along edges if $v_i \notin \vec{v}$, which means $v_a$ could not attain $v_b$ without passing through $v_i$, and vise versa.*

First we should note since $t > 1$, there are at least 2 nodes in $G_s(s_t)$. Suppose $v_i$ is not a cut vertex at $s_t$, then based on Definition 6, $\forall v_a, v_b \in G_s(s_t)$ such that $\exists \vec{v} = (v_a, ..., v_b)$ in which $v_i \notin \vec{v}$. Suppose $a_t = \{v_j\}^+$. Since there is no node deleted, we have $\forall v_a, v_b \in G_s(s_{t+1})/\{v_i, v_j\}, \exists \vec{v} = (v_a, ..., v_b)$ in which $v_i \notin \vec{v}$ based on the connectivity of $G_s(s_t)$. If $v_i$ becomes a cut vertex at $s_{t+1}$, we could have $\forall v_a \in G_s(s_{t+1}), v_a \neq v_i, v_a \neq v_j$ such that $\not\exists \vec{v} = (v_a, ..., v_j)$ where $v_i \notin \vec{v}$. Thus we have $\{v_i, v_j\} \in \mathcal{E}$. If $|\mathcal{N}(a_t)| > 1$, it means $\exists v_k \in G_s(s_{t+1}), v_k \neq v_i, \{v_k, v_j\} \in \mathcal{E}$, it is easy to consider there is a vertex sequence $\vec{v} = (v_a, ..., v_k, v_j)$ where $v_i \notin \vec{v}$, thus $v_i$ is not a cut vertex based on Definition 6. Thus if $v_i$ becomes a cut vertex at $s_{t+1}$, we have $\{v_i, v_j\} \in \mathcal{E}, |\mathcal{N}(a_t)| = 1$.

If $\{v_i, v_j\} \in \mathcal{E}$ and $|\mathcal{N}(a_t)| = 1$, then $\forall v_a \in G_s(s_{t+1}), v_a \neq v_i, v_a \neq v_j$, we have $\{v_a, v_j\} \notin \mathcal{E}$. Therefore, there is no sequence like $\vec{v} = (v_a, ..., v_j)$ without $v_i$. Then based on Definition 6 above, we have $v_i$ is a cut vertex.

Then we complete the proof that if $v_i$ is not a cut vertex at $s_t$, $v_i$ becomes a cut vertex from $s_{t+1}$ iff $|\mathcal{N}(a_t)| = 1$ and $\{v_i, v_j\} \in \mathcal{E}$, where $t > 1$. Based on this, we can get the following Corollary 1.

**Corollary 1** *Suppose an action $a_t = \{v_i\}^+$, there are no cut vertex at $s_t$. If $|\mathcal{N}(a_t)| > 1$, then there are no cut vertex at $s_{t+1}$.*

Next we prove that if $v_i$ is a cut vertex at $s_t$, $v_i$ is not a cut vertex from $s_{t+1}$ iff $|\mathcal{N}(a_t)| > 1$ and $a_t$ connects to all "child groups" of the $v_i$.

We can consider if $v_i$ is a cut vertex at $s_t$, it looks like a parent node in a tree and there are some children of $v_i$. Then if $|\mathcal{N}(a_t)| = 1, \{v_i, v_j\} \in \mathcal{E}$, $v_j$ becomes a new child of $v_i$ in a tree, and thus there is a new "child group" of the $v_i$. Thus if an action $a_t = \{v_k\}^+$ could connect all "child groups" of the $v_i$, then these nodes could attain each other with passing through $v_k$ instead of $v_i$, and thus $v_i$ becomes a non-cut vertex from $s_{t+1}$.

### A.2    PROOF OF THEOREM 1

Next we prove the Theorem 1 by mathematical induction.

1) We first prove that $t = 2$, $v_i$ is a cut vertex at $s_{t+1}$ iff $\exists \mathcal{Z}_{i,j}(s_{t+1}) \neq 0$.

Suppose $t = 2$, then $\mathcal{Z}(s_t) = [\mathbf{0}]_{2\times2}$. Since there are only two nodes $v_a, v_b$ in $G_s(s_t)$, without loss of generality, we define $a_0 = \{v_a\}^+$ and $a_1 = \{v_b\}^+$. Suppose $a_t = \{v_i\}^+$.

If $|\mathcal{N}(a_t)| = 1$, for example, $\{v_a, v_i\} \in \mathcal{E}$ ($v_b$ is symmetrical), then $v_a$ becomes a cut vertex according to Lemma 1. Based on equation 14, equation 16 we have $z(a_t) = [1\ 0]^T$ and

$$\mathcal{Z}'(s_t) = (1 - I_{2\times2}) \wedge \{\mathcal{Z}(s_t) + z(a_t) \cdot [\mathbf{1}]_{1\times2}\} = \begin{bmatrix} 0 & 1 \\ 0 & 0 \end{bmatrix}$$

$$z'(a_t) = \mathcal{Z}'(s_t) \cdot z(a_t) + z(a_t) \wedge \begin{bmatrix} 2 \\ 1 \end{bmatrix} = \begin{bmatrix} 2 \\ 0 \end{bmatrix}$$

Combine these two parts based on equation 15, we have $\mathcal{Z}(s_{t+1}) = \begin{bmatrix} 0 & 1 & 2 \\ 0 & 0 & 0 \\ 0 & 0 & 0 \end{bmatrix}$

Since $\exists \mathcal{Z}_0(s_{t+1}) \neq [\mathbf{0}]_{1\times(t+1)}$, we have $v_a$ becomes a cut vertex, and $v_a$ has two "child groups". Next we prove this by contradiction. Suppose $|\mathcal{N}(a_t)| = 1, \{v_a, v_i\} \in \mathcal{E}$ and $\mathcal{Z}_0(s_{t+1}) = [\mathbf{0}]_{1\times(t+1)}$. Then based on equations above, we have $\mathcal{Z}'_0(s_t) = [\mathbf{0}]_{1\times2}$ and $z'_0(a_t) = 0$.

If $\mathcal{Z}_0'(s_t) = [\mathbf{0}]_{1\times 2}$, since $(1 - I_{2\times 2}) = \begin{bmatrix} 0 & 1 \\ 1 & 0 \end{bmatrix}$, $\mathcal{Z}(s_t) = \begin{bmatrix} 0 & 0 \\ 0 & 0 \end{bmatrix}$, we have $z(a_t) \cdot [\mathbf{1}]_{1\times t} = \begin{bmatrix} 0 & 0 \\ 0 & 0 \end{bmatrix}$, thus $z(a_t) = [0\ 0]^T$, which contradicts to our statement that $z(a_t) = [1\ 0]^T$. Thus we prove that $t = 2$, if $|\mathcal{N}(a_t)| = 1$, then $v_i$ is a cut vertex at $s_{t+1}$ iff $\exists \mathcal{Z}_{i,j}(s_{t+1}) \neq 0$.

If $|\mathcal{N}(a_t)| > 1$, since there are only 2 nodes in $G_s(s_t)$, thus we have $|\mathcal{N}(a_t)| = 2$, $\{v_a, v_i\} \in \mathcal{E}$ and $\{v_b, v_i\} \in \mathcal{E}$. Then based on Corollary 1, there are no cut vertices in $s_{t+1}$. Based on equation 14,equation 16 we have $z(a_t) = [1\ 1]^T$ and

$$\mathcal{Z}'(s_t) = \begin{bmatrix} 0 & 0 \\ 0 & 0 \end{bmatrix}, \qquad\qquad z'(a_t) = \begin{bmatrix} 0 \\ 0 \end{bmatrix},$$

where $\mathbb{I}_{set} = 0$ for both $\mathcal{Z}_0(s_t)$ and $\mathcal{Z}_1(s_t)$, since $set(\mathcal{Z}_0(s_t) \wedge [1\ 1]) = set(\mathcal{Z}_0(s_t)) = [0]$, $set(\mathcal{Z}_1(s_t) \wedge [1\ 1]) = set(\mathcal{Z}_1(s_t)) = [0]$. $\mathbb{I}_{sum} = 0$ since $\sum \mathcal{Z}'(s_t)z(a_t) = 0$.

Combine these two parts based on equation 15, we have $\mathcal{Z}(s_{t+1}) = \begin{bmatrix} 0 & 0 & 0 \\ 0 & 0 & 0 \\ 0 & 0 & 0 \end{bmatrix}$.

Next we prove this by contradiction. Suppose $|\mathcal{N}(a_t)| = 2, \{v_a, v_i\} \in \mathcal{E}, \{v_b, v_i\} \in \mathcal{E}$ and $\exists \mathcal{Z}_{0,j}(s_{t+1}) \neq 0, j = 0, 1, 2$, then we have three cases as follows,

- If $\mathcal{Z}_{0,0}'(s_t) \neq 0$, which contradicts to equation 16 since $(1 - I_{2\times 2}) = \begin{bmatrix} 0 & 1 \\ 1 & 0 \end{bmatrix}$.

- If $\mathcal{Z}_{0,1}'(s_t) \neq 0$, then we have $\mathbb{I}_{set} = 1$, which corresponds to $set(\mathcal{Z}_0(s_t) \wedge z^T(a_t)) \neq set(\mathcal{Z}_0(s_t))$. However, $set(\mathcal{Z}_0(s_t) \wedge z^T(a_t)) = set(\mathcal{Z}_0(s_t)) = [0]$ since $\mathcal{Z}_0 = [\mathbf{0}]_{1\times 2}$, thus it contradicts to the statement.

- If $z_0'(a_t) \neq 0$, then we have $\mathbb{I}_{sum} = 1$, which means $\sum \mathcal{Z}_0'(s_t)z(a_t) \neq 0$, since $z(a_t) = [1\ 1]^T$, then we should have $\mathcal{Z}_0'(s_t) \neq [\mathbf{0}]_{1\times 2}$, which contradicts to the cases above.

Thus we prove that $t = 2$, if $|\mathcal{N}(a_t)| > 1$, then $v_i$ is a cut vertex at $s_{t+1}$ iff $\exists \mathcal{Z}_{i,j}(s_{t+1}) \neq 0$.

Above all, for $t = 2$, we have proved that $v_i$ is a cut vertex at $s_{t+1}$ iff $\exists \mathcal{Z}_{i,j}(s_{t+1}) \neq 0$.

2) Next we consider $t > 2$, suppose at $s_t$, we have $v_i$ is a cut vertex at $s_t$ iff $\exists \mathcal{Z}_{i,k}(s_t) \neq 0, k \neq i$. Suppose $a_t = \{v_j\}^+$. We need to prove that $v_i$ is a cut vertex at $s_{t+1}$ iff $\exists \mathcal{Z}_{i,k}(s_{t+1}) \neq 0, k \neq i$.

If $|\mathcal{N}(a_t)| = 1$, without loss of generality, we consider $\{v_i, v_j\} \in \mathcal{E}, v_i \in G_s(s_t)$, then based on equation 14, we have $z(a_t) = [0 \cdots 1 \cdots 0]^T$, where $z_i(a_t) = 1, z_k(a_t) = 0, \forall k \neq i$.

If $v_i$ is not a cut vertex, we have $\mathcal{Z}_i(s_t) = [\mathbf{0}]_{1\times t}$. If $v_i$ is a cut vertex, we have $\mathcal{Z}_{i,k}(s_t) \neq 0, \forall k \neq i$. $\mathcal{Z}_i(s_t)$ is the row corresponding to $v_i$.
Then based on equation 16, we have

$$\mathcal{Z}_i'(s_t) = (1 - I_{t\times t})_i \wedge \{\mathcal{Z}(s_t) + \mathbb{I}_{\mathcal{Z}_i(s_t)[1-z(a_t)]=0}\, z(a_t) \cdot [\mathbf{1}]_{1\times t}\}_i.$$

We should check $\mathbb{I}_{\mathcal{Z}_i(s_t)[1-z(a_t)]=0}$ and there two different cases. Before giving the proof, we introduce Lemma 2 as follows,

**Lemma 2** *Suppose $a_t = \{v_j\}^+, \mathcal{N}(a_t) = 1, v_i \in G_s(s_t)$. If $\mathcal{Z}_i(s_t)[1 - z(a_t)] \neq 0$, $v_j$ connects to a cut vertex $v_i$ , and if $\mathcal{Z}_i(s_t)[1 - z(a_t)] = 0$, $v_j$ connects to a non-cut vertex $v_i$.*

a) If $\mathcal{Z}_i(s_t)[1 - z(a_t)] = 0$, which means $v_j$ connects to a non-cut vertex based on Lemma 2. If $v_i$ is not a cut vertex at $s_t$, we have $\mathcal{Z}_i(s_t) = [\mathbf{0}]_{1\times t}$ and

$$\mathcal{Z}_i'(s_t) = [1 - I_{t\times t}]_i \wedge \{\mathcal{Z}_i(s_t) + [z(a_t) \cdot [\mathbf{1}]_{1\times t}]_i\} = [1 \cdots 0 \cdots 1].$$

where $\mathcal{Z}_{i,i}'(s_t) = 0, \mathcal{Z}_{i,k}'(s_t) = 1, \forall k \neq i$. And we have

$$z_i'(a_t) = [\mathcal{Z}'(s_t) \cdot z(a_t)]_i + z_i(a_t) \wedge [\max\{\mathcal{Z}_i'(s_t)\} + 1] = 2$$

Combine these two parts based on equation 15, we have $\mathcal{Z}_i(s_{t+1}) = [1 \cdots 0 \cdots 1\ 2]$, where $\mathcal{Z}_{i,i}(s_{t+1}) = 0, \mathcal{Z}_{i,k}(s_{t+1}) = 1, k \neq i, k \leq t - 1, \mathcal{Z}_{i,t}(s_{t+1}) = 2$. Therefore we have

$\exists \mathcal{Z}_{i,k}(s_{t+1}) \neq 0, k = 0, ..., t$. Since $v_i$ is not a cut vertex, $|\mathcal{N}(a_t)| = 1$, $\{v_i, v_j\} \in \mathcal{E}$, based on Lemma 1 we know $v_i$ is a cut vertex at $s_{t+1}$.

b) If $\mathcal{Z}_i(s_t)[1 - z(a_t)] \neq 0$, which means there are some some cut vertices at $s_t$, and $v_j$ connects to a cut vertex based on Lemma 2. In this case, it is easy to consider $v_i$ will still be a cut vertex at state $s_{t+1}$ since $\forall v_k \in G_s(s_{t+1}), v_k \neq v_j, v_k \neq v_i, v_k$ can not attain $v_j$ along edges without passing through $v_i$ based on $|\mathcal{N}(a_t)| = 1$, and vice versa.
Without loss of generality, suppose $\mathcal{Z}_{i,i}(s_t) = 0, \mathcal{Z}_{i,k}(s_t) = 1, k < t, k \neq i, \mathcal{Z}_{i,t}(s_t) = 2$. This assumption is the case that $a_{t-1} = \{v_a\}^+, |\mathcal{N}(a_{t-1})| = 1, v_i$ becomes a cut vertex at $s_t$. Then based on equation 16, we have

$$\mathcal{Z}'_i(s_t) = (1 - I_{t \times t})_i \wedge \mathcal{Z}_i(s_t) = \mathcal{Z}_i(s_t)$$
$$z'_i(a_t) = \mathcal{Z}'_i(s_t) \cdot z(a_t) + z_i(a_t) \wedge [\max\{\mathcal{Z}'_i(s_t)\} + 1] = 3$$

Combine these two parts based on equation 15, we have $\mathcal{Z}_j(s_{t+1}) = [1 \cdots 0 \cdots 2 \ 3]_{1 \times (t+1)}$. It is easy to see $v_j$ could divide $G_s(s_{t+1})$ into 3 parts.
Thus we have shown that if $|\mathcal{N}(a_t)| = 1$, $v_i$ is a cut vertex iff $\exists \mathcal{Z}_{i,k}(s_t) \neq 0, k \neq i$.

If $|\mathcal{N}(a_t)| > 1$, without loss of generality, we can suppose $|\mathcal{N}(a_t)| = k, k \geq 2$, then we have $k$ non-zero positions in $z(a_t)$. Before giving the proof, we propose Lemma 3 as follows,

**Lemma 3** Suppose $a_t = \{v_j\}^+$, $v_i$ is a cut vertex at $s_t$, then $v_i$ becomes a non-cut vertex at $s_{t+1}$ iif $set(\mathcal{Z}_i(s_t) \wedge z(a_t)) = set(\mathcal{Z}_i(s_t))$.

Suppose $v_i$ is a cut vertex in $s_t$, which means $\forall k \neq i, \mathcal{Z}_{i,k}(s_t) \neq 0, \mathcal{Z}_i(s_t)$ has $t - 1$ non-zero positions. We need to check whether $v_j$ connects to all "child groups" of $v_i$, then $v_i$ may not be a cut vertex after adding $v_j$. Then there are following two cases:

a) If $set(\mathcal{Z}_i(s_t) \wedge z(a_t)) = set(\mathcal{Z}_i(s_t)) \neq [0]$, then based on Lemma 3 we have $v_i$ becomes a non-cut vertex at $s_{t+1}$. Based on equation 17 we have

$$\mathcal{Z}'_i(s_t) = [\mathbf{0}]_{1 \times t} \qquad z'_i(a_t) = 0$$

Then based on equation 16 we have $\mathcal{Z}_j(s_{t+1}) = [\mathbf{0}]_{1 \times (t+1)}$

b) $set(\mathcal{Z}_i(s_t) \wedge z(a_t)) \neq set(\mathcal{Z}_i(s_t))$, then based on Lemma 3 we have $v_i$ is still a cut vertex at $s_{t+1}$. Based on equation 17 we have

$$\mathcal{Z}'_{i,k}(s_t) = \mathbb{I}_{set2} \max\{set(\mathcal{Z}_i(s_t) \wedge z^T(a_t))\} + (1 - \mathbb{I}_{set2}) \mathcal{Z}_{i,k}(s_t)$$

Since $v_i$ is a cut vertex in $s_t$, thus $\mathcal{Z}_{i,k}(s_t) \neq 0, \forall i \neq k$. Since $|\mathcal{N}(a_t)| > 1$, we have $\mathcal{Z}_i(s_t) \wedge z(a_t) \neq [\mathbf{0}]_{1 \times t}$. Therefore, it is easy to know $\exists \mathcal{Z}'_{i,k}(s_t) \neq 0$. Then based on equation 16 we have $\exists \mathcal{Z}_{i,k}(s_{t+1}) \neq 0, k \neq i$.

Above all, for $t > 2$, we have proved that $v_i$ is a cut vertex at $s_{t+1}$ iff $\exists \mathcal{Z}_{i,j}(s_{t+1}) \neq 0$.

Therefore, we complete the proof that for $t \geq 2$, $v_i$ is a cut vertex at $s_t$ iff $\exists \mathcal{Z}_{i,j}(s_t) \neq 0$.

### A.3    PROOF OF LEMMA 2

If $\mathcal{N}(a_t) = 1$ and $\{v_j, v_i\} \in \mathcal{E}$, $z(a_t) = [0...1...0]^T$, where $z_i(a_t) = 1$ and $z_k(a_t) = 0, k \neq i$.

If $v_i$ is a cut vertex, we have $\mathcal{Z}_{i,i}(s_t) = 0$ and $\mathcal{Z}_{i,k}(s_t) \neq 0$, where $k \neq i$. Then based on the precise matrix multiplication, we should have $\mathcal{Z}_i(s_t)[1 - z(a_t)] \neq 0$.

If $v_i$ is not a cut vertex, then $\mathcal{Z}_i(s_t) = [\mathbf{0}]_{1 \times t}$ and we should have $\mathcal{Z}_i(s_t)[1 - z(a_t)] = 0$

### A.4    PROOF OF LEMMA 3

Since $v_i$ is a cut vertex at $s_t$, then $set(\mathcal{Z}_i(s_t))$ has at least 2 different values based on calculations before, since we define $set(\cdot)$ contains distinct value except of 0.

If $|\mathcal{N}(a_t)| = 1$, which means $z(a_t)$ has only one non-zero position, it is easy to consider $v_i$ is still a cut vertex at $s_{t+1}$. If $z_i(a_t) = 1$, which means $\{v_j, v_i\} \in \mathcal{E}$, then $set(\mathcal{Z}_i(s_t) \wedge z(a_t)) = \emptyset \neq$

$set(\mathcal{Z}_i(s_t))$. If $z_k(a_t) = 1, k \neq i$, then $set(\mathcal{Z}_i(s_t) \wedge z(a_t)) = [\mathcal{Z}_{i,k}(s_t)] \neq set(\mathcal{Z}_i(s_t))$. Thus if $|\mathcal{N}(a_t)| = 1$, we have $set(\mathcal{Z}_i(s_t) \wedge z(a_t)) \neq set(\mathcal{Z}_i(s_t))$ and $v_i$ is still a cut vertex at $s_{t+1}$.

Based on Lemma 1 we know only $|\mathcal{N}(a_t)| = 1$ will potentially introduce a new cut vertex, or add a new "child group" for the current cut vertex. If $a_t$ connects to all "child groups" of a cut vertex $v_i$, then $v_i$ becomes a non-cut vertex from $s_{t+1}$.

If $|\mathcal{N}(a_t)| = m, m > 1$. Since $z(a_t)$ has $m$ non-zero positions, then $\mathcal{Z}_i(s_t) \wedge z(a_t)$ might have 0, $m - 1$ or $m$ non-zero positions as following different cases:

- If $\mathcal{Z}_i(s_t) \wedge z(a_t)$ are all zeros, $z_i$ is not a cut vertex.
- If $\mathcal{Z}_i(s_t) \wedge z(a_t)$ has $m - 1$ non-zero positions, $v_i$ is a cut vertex and $\{v_i, v_j\} \in \mathcal{E}$.
- If $\mathcal{Z}_i(s_t) \wedge z(a_t)$ has $m$ non-zero positions, $v_i$ is a cut vertex and $\{v_i, v_j\} \notin \mathcal{E}$.

Without loss of generality, we start with the case $\mathcal{Z}_i(s_t) = [1 \cdots 0 \cdots 1 \ 2]$, which means $a_{t-1} = \{v_a\}^+$ makes $v_i$ become a cut vertex at $s_t$. Thus $\mathcal{Z}_{i,k}(s_t) = 1, k < t, k \neq i, \mathcal{Z}_{i,t}(s_t) = 2, \mathcal{Z}_{i,i}(s_t) = 0$. Then we have $v_i$ has two "child groups", the first group consists of all nodes in $G_s(s_t)$ except of $v_i$, the second group only has one node $v_a$. Suppose $a_t = \{v_j\}^+$ and $|\mathcal{N}(a_t)| = m$.

- If $m = 2, \{v_a, v_j\} \in \mathcal{E}, \{v_i, v_j\} \in \mathcal{E}$, then $z_i(a_t) = z_t(a_t) = 1$ and we have $set(\mathcal{Z}_i(s_t) \wedge z(a_t)) = [2]$ while $set(\mathcal{Z}_i(s_t)) = [1, 2]$. Based on Lemma 1 we know $a_t$ does not connect two "child groups" and thus $v_i$ is a cut vertex at $s_{t+1}$.
- If $m = 2, \{v_a, v_j\} \in \mathcal{E}$ and $\{v_k, v_j\} \in \mathcal{E}, v_k \neq v_i$ then $z_k(a_t) = z_t(a_t) = 1, z_i(a_t) = 0$ and we have $set(\mathcal{Z}_i(s_t) \wedge z(a_t)) = [1, 2] = set(\mathcal{Z}_i(s_t))$. Based on Lemma 1 we know $a_t$ connect two "child groups" and thus $v_i$ is not a cut vertex at $s_{t+1}$.
- If $m > 2, \{v_a, v_j\} \notin \mathcal{E}$, we can easily get $set(\mathcal{Z}_i(s_t) \wedge z(a_t)) = [1] \neq set(\mathcal{Z}_i(s_t))$. Based on Lemma 1 we know $a_t$ only connects $v_a$ and thus $v_i$ is a cut vertex at $s_{t+1}$.

Above all, we complete the proof for Lemma 3.

## B  DISCUSSIONS

### B.1  PARENT EXPLORATIONS IN DAG MATTERS

Since graph $G_s(s_t)$ is generated by sequential actions, the trajectory becomes an ordered node sequence. However, we should note that the generated subgraph should be an unordered set, which means it is independent of the sequence but determined by the connectivity of the nodes. For example, for a graph consisting of three nodes, if there are pair-wise edges between these three nodes, the generated graph will be the same regardless of the order of nodes. However, if only two edges connect these three nodes, then the intermediate node as a bridge cannot be the last one added. We can conclude that **the sequence matters when the ordering of adding nodes will affect the connectivity of a graph.**

There may be many trajectories that lead to the same state $s_t$, while sampling a single trajectory $\tau$ each time could not contain this information. In order to solve this problem, RGExplainer Shan et al. (2021) applied the pre-training strategies with maximum Log-Likelihood Estimation (MLE) over all possible generated orderings for an explanatory graph Vinyals et al. (2015). In contrast, our GFlowExplainer is modeled based on a directed acyclic graph structure, as multiple action sequences lead to the same graph, and the direct parent explorations for flow matching conditions "connect" these trajectories together, which naturally eliminates the influence of orderings. Therefore, there is no need to pretrain, and we can learn the policy to generate good enough candidate graphs.

We also conduct experiments to show the importance of connectivity constraints for loss convergence in Appendix D.2.

## C  ALGORITHMS

We show the pseudocode of our GFlowExplainer for node classification and graph classification in Algorithm 1 and Algorithm 2 respectively.

For node classification tasks, given an input graph $G = (\mathcal{V}, \mathcal{E})$ and its features $\mathcal{X}$, a trained GNN model $\Phi$ and node instances $\mathcal{I}$, GFlowExplainer aims to train a generative policy $\pi(a_t \mid s_t)$ and find the explanatory subgraph $G_s^{(i)}$ for $i$-th node instance. Considering the prediction of a node instance is determined by its $L$-hop neighborhoods based on the message passing scheme in GNNs, in which $L$ is the number of layers in the trained model $\Phi$.

During each training epoch, GFlowExplainer parallel generates $s_f$ for each node $v_i \in \mathcal{I}$ based on policy $\pi(a_t \mid s_t)$ by sequential actions. For every iteration, GFlowExplainer samples a valid action $a_t : \{v_j\}^+ \sim \pi(a \mid s_t)$ s.t. $v_j \in \mathcal{N}(s_t)$ based on the generative flow network to make a state transition $s_t \rightarrow s_{t+1}$ and explore valid parents based on the updated cut vertex matrix $\mathcal{Z}(s_{t+1})$. The terminal state $s_f$ is generated once the stopping criteria is reached. When the epoch number $E$ is reached, the trained policy $\pi(a_t \mid s_t)$ based on the flow matching loss generates explanatory subgraphs in the inference time for evaluation.

As for the graph classification task, the difference lies in the choice of starting node. Therefore GFlowExplainer need to train an additional locator $\mathbb{L}$ during the training process. The final graph representations $z_{g_n}$ and node representations $z_{v_i,n}$ are computed based on the trained GNN model $\Phi$. Then $\mathbb{L}$ is trained with policy $\pi$ coordinately.

---

**Algorithm 1** GFlowExplainer for node classification

**Require:** $G = (\mathcal{V}, \mathcal{E})$: Graph ; $\mathcal{X}$: Node features ; $\mathcal{I}$: Node instances ; $B$: batch size ; $E$: epoch number ; $\eta$: learning rate ; $\Phi$: trained GNN classification model
1: **repeat**
2:      **repeat** *(For each node $v_i \in \mathcal{I}$, parallel do with a batch size $B$)*
3:          Initialize $s_0 = \{v_i\}$, $\mathcal{Z}(s_0) = I_{2 \times 2}$
4:          Construct $X_t'$, $\bar{H}_t^L$ according to equation 7, equation 8 and equation 9
5:          Sample a valid action $a_t : \{v_j\}^+ \sim \pi(a|s_t)$ s.t. $v_j \in \mathcal{N}(s_t)$
6:          Make a state transition $s_{t+1} = s_t \cup \{v_j\}$
7:          Update $\mathcal{Z}_{t+1}$ according to equation 14
8:          Explore all valid parents with $(s_p, a_p)$ based on $\mathcal{Z}_{t+1}$
9:      **until** Attain the stopping criteria
10:      Calculate $r(s_f, Y)$ based on equation 10
11:      Update the parameters $\{\Theta_1, \Theta_2, \theta_1\}$ based on $\nabla \mathcal{L}(\tau)$ and $\eta$
12: **until** epoch number $E$ is reached
**Ensure:** Policy $\pi(a_t \mid s_t)$ and generated explanatory subgraph $G_s^{(i)}$ during the inference phase

---

**Algorithm 2** GFlowExplainer for graph classification

**Require:** $G^{(n)} \in \mathcal{I}$: Graph instances ; $B$: batch size ; $E$: epoch number ; $\eta$: learning rate ; $\Phi$: trained GNN classification model
1: **repeat**
2:      **repeat** *(For each node $G^{(n)} \in \mathcal{I}$, parallel do with a batch size $B$)*
3:          Initialize $s_0 = \{\mathbb{L}(G^{(n)})\}$, $\mathcal{Z}(s_0) = I_{2 \times 2}$
4:          Construct $X_t'$, $\bar{H}_t^L$ according to equation 7, equation 8 and equation 9
5:          Sample a valid action $a_t : \{v_j\}^+ \sim \pi(a|s_t)$ s.t. $v_j \in \mathcal{N}(s_t)$
6:          Make a state transition $s_{t+1} = s_t \cup \{v_j\}$
7:          Update $\mathcal{Z}_{t+1}$ according to equation 14
8:          Explore all valid parents with $(s_p, a_p)$ based on $\mathcal{Z}_{t+1}$
9:      **until** Attain the stopping criteria
10:      Calculate $r(s_f, Y)$ based on equation 10
11:      Update the parameters $\{\Theta_1, \Theta_2, \theta_1\}$ based on $\nabla \mathcal{L}(\tau)$ and $\eta$
12:      # *Coordinate train $\mathbb{L}$*
13:      **for** each sampled graph $G^n \in \mathcal{I}$ **do**
14:          $D = [\pi(s_f \mid v_i), -\mathcal{L}(\pi(s_f \mid v_i))]$
15:          Update parameters in $\mathbb{L}$ on $D$
16:      **end for**
17: **until** epoch number $E$ is reached
**Ensure:** Policy $\pi(s_f \mid \mathbb{L}(G^{(n)}))$ and generated explanatory subgraph $G_s^{(i)}$ during the inference phase

Table 2: Inference Time

| Inference Time | BA-Shapes | BA-Community | Tree-Cycles | Tree-Grid | BA-2motifs | MUTAG |
|---|---|---|---|---|---|---|
| GNNExplainer | 53ms | 76ms | 49ms | 57ms | 34ms | 16ms |
| PGExplainer | 12ms | 17ms | 3ms | 2ms | 1ms | 14ms |
| RGExplainer | 8ms | 2ms | 7ms | 6ms | 5ms | 9ms |
| Ours | 7ms | 3ms | 8ms | 11ms | 6ms | 7ms |

Table 3: Training Time

| | Training Time | BA-Shapes | BA-Community | Tree-Cycles | Tree-Grid | BA-2motifs | MUTAG |
|---|---|---|---|---|---|---|---|
| Pretrining | RGExp | 102.56s | 143.73s | 83.36s | 463.61s | 1294.26s | 2918.22s |
| | Ours | – | – | – | – | – | – |
| Update per epoch | RGExp | 24s | 22s | 11s | 5s | 14s | 28s |
| | Ours | 8s | 10s | 9s | 10s | 7s | 12s |

# D   ADDITIONAL RESULTS

## D.1   EFFICIENCY ANALYSIS

We also compare the inference time of GNNExplainer, PGExplainer, RGExplainer, and our GFlow-Explainer with the same environment. We compute the average inference time for explaining a single instance for each task and report the results in Table 2. We could find that GNNExplainer is the slowest, and the inference time of RGExplainer, PGExplainer and our GFlowExplainer are in the same order of magnitude. Therefore, we can conclude that the GFlowNets-based framework will not require a longer inference time.

Since the sampling procedure for a connected subgraph is similar between RGExplainer and GFlow-Explainer. We also report the training time of our GFlowExplainer and compare it with the RGExplainer, whose pre-training part is also included. The comparison results are shown in Table 3. As we mentioned before, GFlowExplainer does not need pre-training process. However, we can find that pre-training strategies of RGExplainer take much time and even become dominating in the total running time. As for the time of iterative update per epoch, GFlowExplainer is overall faster than RGExplainer, which could also show the efficiency of the proposed Theorem 1 for updating cut vertices. The GFlowExplainer is more practical than other learning-based approaches for large-scale datasets.

## D.2   LOSS CONVERGENCE ANALYSIS

In this section we conduct more ablation experiments to show the role of connectivity constraints for flow loss convergence. In the previous ablation experiments (refer to Section 4.4), we compare GFlow-Sequence and GFlowExplainer on the explanation performance in the inductive setting. In this section we consider add DAG structures without connectivity constraints, that is, there are $(|G_s(s_t)| - 1)$ direct parents ( because the node to be interpreted could not be deleted, and $|G_s(s_t)|$ corresponds to the number of nodes in the subgraph ) for state $s_t$. We call this approach is GFlow-Graph. Based on the theoretical sense, breaking connectivity constraints while exploring parent states will make the inconsistency between action space and trajectories in the DAG structure. We visualize the flow matching loss of both GFlowExplainer and GFlow-Graph. We set 5 different seeds on the BA-shape datasets, 16 batches with 80 epochs for each sampling.

Figure 4 shows the flow matching loss of GFlowExplainer and GFlow-Graph. Since the original flow loss is small at the beginning of training, we expand the multiples of the regular items in the reward function to make the loss relatively high. As a result, we can find that both approaches could attain convergence, but GFlowExplainer has lower losses, which confirms our statement. Furthermore, both approaches could converge fast because once we confirm the starting point, the allowed action space reduces significantly due to the connectivity constraints and the stopping criteria, thus making the flow calculation easier. We also plot convergence analysis for other datasets in Figure 5. We

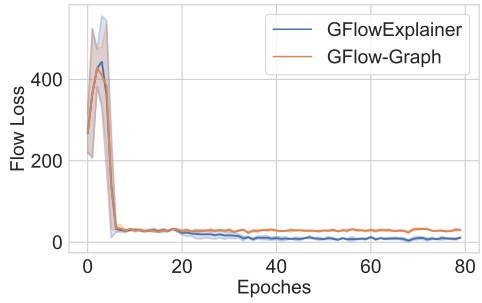

Figure 4: Flow matching loss for GFlowExplainer and GFlow-Graph

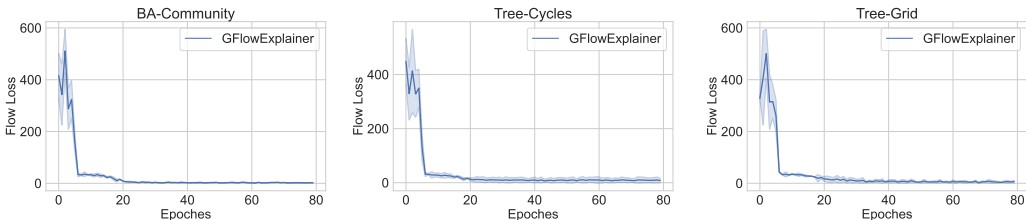

Figure 5: Convergence Analysis

experimentally observe that GFlowExplainer could converge on all these datasets, where the x-axis represents the training epoch, and the y-axis indicates the flow matching loss.

### D.3   COMPUTATIONAL ANALYSIS OF THEOREM 1

Tarjan's strongly connected components algorithm Tarjan (1972) need to iterate all nodes and edges of the subgraph $G_s(s_t)$. The time complexity is $\mathcal{O}(|\mathcal{V}| + |\mathcal{E}|)$ for each state, which is inefficient for dynamic graphs. In addition, exploring all edges and nodes is a disaster for space complexity with a large graph and is not applicable in real-world applications. However, we can update and store the cut vertex without repeatedly checking by taking advantage of the step-by-step generative process. The idea is to snap to the properties of a cut vertex in dynamic graphs and identify conditions for transformations between cut and non-cut vertices.

To show the effectiveness and efficiency of the proposed Theorem 1, in this section, we visualize the cut vertices in dynamic graphs via a simple simulation experiment. In addition, we show the time comparison between our proposed algorithm with other traditional cut vertices algorithms.

We construct a $10 \times 10$ adjacency matrix to represent an undirected connected graph, and starting with two nodes; the subgraph adds a neighbor node sequentially. We record the time of the Tarjan's algorithm and our approach. For a fair comparison, the updating process time in our method is also included. We report the accumulated time in Figure 6. We can find that with the increasing size of the graph, the accumulated time of Tarjan's algorithm increases sharply while our approach increases linearly. We also visualize the cut vertices exploration process in dynamic graphs in Figure 7, in which the black dot represents the action of adding that node, pink dots correspond to the cut vertex, and the orange dots are regular nodes in the subgraph. It is easy to find that only the action node will introduce a new cut vertex or delete a cut vertex. Therefore, we can only iterate the new edges introduced by the action node and check its connectivity relationships with other nodes in the subgraph. In contrast, Tarjan's algorithm will iterate all nodes and edges in the subgraph after adding the action node; thus, it makes sense that our approach has smaller time complexity.

### D.4   MORE INDUCTIVE SETTING RESULTS

Due to space limit, we add some inductive experiments in this section. The Figure 8 shows the inductive experiments of four algorithms on Tree-Grid and MUTAG Datasets. As for the reinforcement learning based approches, the performance of Tree-Grid is similar to that of Tree-Cycles. Without pre-training strategies, the AUC value of RG-NoPretrain remains at 0.5 and RGExplainer could provide better explanations with the increasing size of the training instances. The ratio change of the

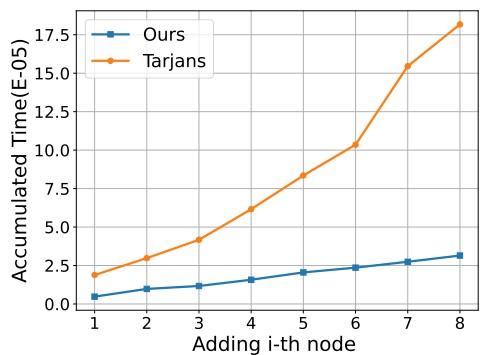

Figure 6: Time Comparison of exploring cut vertices

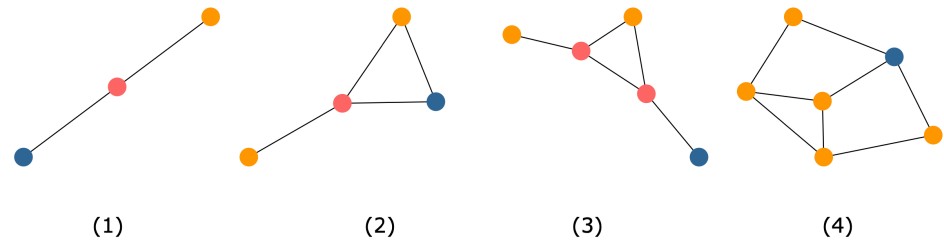

Figure 7: Cut vertices exploration in dynamic graphs: From left to right is the process of adding nodes step-by-step. It is easy to find that only the action node (black) will introduce a new cut vertex (pink) or delete a cut vertex.

training set has more significant influence on the performance of GFlow-Sequence. GFlowexplainer can always maintain high performance, showing its strong generalization ability. In the MUTAG dataset, both GFlowExplainer and RGExplainer perform well.

## E  IMPLEMENTATION DETAILS

### E.1  BASELINES

- GNNExplainer Ying et al. (2019) is the first formal approach to explain trained GNNs, which defines the problem as an optimization task to maximize the mutual information between the predicted labels and the distribution of possible subgraphs with some constraints. Codes are available at https://github.com/RexYing/gnn-model-explainer

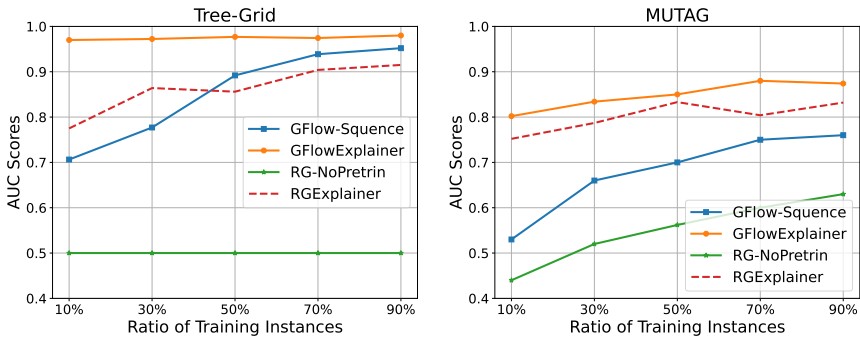

Figure 8: Inductive setting with ablation experiments on Tree-Grid Dataset and MUTAG Dataset

Table 4: Hyper-parameters

| Hyper-parameters | Value |
|---|---|
| Batch Size | 64 |
| Number of layers of APPNP | 3 |
| $\alpha$ in APPNP | 0.85 |
| Hidden dimension | 64 |
| Architecture of MLP in $\mathbb{L}$ | 64-8-1 |
| Learning rate | 1e-2 |
| Optimizer | Adam |
| Number of hops | 3 |
| Maximum size of generated sequences | 20 |
| Training epochs (node tasks) | {50,100} |
| Training epochs (graph tasks) | 100 |
| Sample ratio of graph instance to train $\mathbb{L}$ | 0.2 |

- PGExplainer Luo et al. (2020) leverages the representations generated by the trained GNN and adopts a deep neural network to learn the crucial nodes and edges. Codes are available at https://github.com/flyingdoog/PGExplainer

- DEGREE Feng et al. (2021) proposes a decomposition-based explanation method for graph neural networks, which directly decomposes the influence of node groups in the forward pass. The decomposition rules are designed for GCN and GAT. Further, to efficiently select subgraph groups from all possible combinations, the authors propose a greedy approach to search for maximally influential node sets. Codes are available at https://github.com/Qizhang-Feng/DEGREE

- RGExplainer Shan et al. (2021) utilises the Reinforcement Learning to generate the instance-level explanations for GNNs. The seed locator and stopping criteria to find the most influential node in a graph instance and check whether the generated explanatory graph are good enough. Codes are available at https://openreview.net/forum?id=nUtLCcV24hL

## E.2 EXPERIMENT ENVIRONMENT

All experiments were conducted on a NVIDIA Quadro RTX 6000 environment with Pytorch. The parameters of GFlowExplainer are shown in Table 4.

## E.3 DETAILS ABOUT DATASET

We show the data statistics in Table 5. In this paper we consider the following five datasets:

- *The BA-shapes data set* consists of one Barabasi-Albert graph Barabási & Albert (1999) as the base and 80 house-structure motifs. Each motif is randomly attached to a node in BA graph and extra edges are added as noises;

- *The BA-community dataset* is comprised of two BA-shapes with different node features generated by Gaussian distributions. The extra edges are also connect two BA-shapes;

- *The Tree-cycles dataset* includes a multi-level binary tree as the base and 80 six-node cycle motifs. The cycle motifs are randomly attached to the tree.

- *The BA-motifs dataset* has 1000 graphs where half of them are a BA graph attached with a house-structure motif, while the rest are a BA graph attached with a five-node cycle motif;

- *The Mutagenicity dataset* is a real dataset, which includes 4337 molecule graphs. They can be classified as mutagenic or nonmutagenic depending on whether having $NH_2$ or $NO_2$ motifs.

Table 5: Dataset statistics

| | **Node Classification** | | | | **Graph Classification** | |
|---|---|---|---|---|---|---|
| | BA-Shapes | BA-Community | Tree-Cycles | Tree-Grid | BA-2motifs | MUTAG |
| #graphs | 1 | 1 | 1 | 1 | 1,000 | 4,337 |
| #nodes | 700 | 1,400 | 871 | 1,231 | 25,000 | 131,488 |
| #edges | 4,110 | 8,920 | 1,950 | 3,410 | 51,392 | 266,894 |
| #labels | 4 | 8 | 2 | 2 | 2 | 2 |

Figure 9: Base and Motif structures for each dataset Ying et al. (2019)

# F    ADDITIONAL EXPERIMENTS

## F.1    TRAJECTORY BALANCE OBJECTIVE

In the trajectory balance loss objective, we need to parameterize the $Z_\theta$, $\mathcal{P}_F(s_{t+1}|s_t, \theta)$ and optional $\mathcal{P}_B(s_t|s_{t+1}, \theta)$. One natural choice of $\mathcal{P}_B(s_t|s_{t+1})$ is to set it to be uniform over all the valid parents of a state $s_{t+1}$, i.e., $\mathcal{P}_B(\cdot|s_{t+1}) = 1/\#\{s_t|(s_t \rightarrow s_{t+1} \in \mathcal{A}\}$ suggested in Malkin et al. (2022). In our case we only parameterize the former two terms.

Previous TB suggests to parameterize $Z_\theta$ with a constant since it considers the unconditional case. Therefore, it only need approximate the total flow $Z$ so that $Z = R(x), \forall x \in \mathcal{X}$. In contrast, our task applies the state-conditional GFlowNets, which means there are various subgraph flows $Z_s$ we need to approximate to get $Z_s = R(x|s)$. Simply speaking, for each $\mathcal{G}_s$, the $Z_s$ should be different.

To show the complexity of trajectory balance in state-conditional GFlowNets, we have two attempts.

First, we follow the unconditional case and parameterize $Z_\theta$ with a constant for initialization. This is the same as to Malkin et al. (2022). Our objective is to learn the parameters $\theta$ of the forward conditional policies $\mathcal{P}_F(s_{t+1}|s_t, v_0; \theta)$ and $\log Z_\theta$.

Second, we consider the conditional flow approximation. Our objective is to learn the parameters $\theta$ of the forward conditional policies $\mathcal{P}_F(s_{t+1}|s_t, v_0; \theta)$ and function $\log Z_\theta(v_0)$. Therefore, $\forall \tau = (s_0, ..., s_{n+1} = s_f) \in \mathcal{T}$, we define the state-conditional trajectory balance as follows,

$$\mathcal{L}(\tau, v_0; \theta) = \left( \log \frac{Z_\theta(v_0) \prod_{s_t \rightarrow s_{t+1} \in \tau} \mathcal{P}_F(s_{t+1}|s_t, v_0; \theta)}{r(s_f|v_0) \prod_{s_t \rightarrow s_{t+1} \in \tau} \mathcal{P}_B(s_t|s_{t+1}, v_0)} \right)^2. \tag{19}$$

In our experiment, we train a three-layer MLP to model $Z_\theta(\cdot)$. The input is the node features of $v_0$ in each trajectory and the output is the approximated flow. The learning rate for $Z_\theta(\cdot)$ is 0.1, the hidden layer size is 128.

We plot the loss convergences for both approaches with datasets BA-Shapes in Figure 10 . The unconditional flow could not converge at all and the AUC maintains $0.5 \sim 0.6$, the conditional flow could converge after some fluctuations, but the AUC has high variances, shown in Table 6.

We guess the reason behind is that in our task, the loss decreases with the change of both $Z_\theta(-)$ and $\mathcal{P}_F(-|-, -; \theta)$. Even though the loss could converge, without good approximation of $Z_\theta(-)$,

Table 6: Comparisons among Flow Matching, Trajectory Balance (unconditional-$Z$) and Trajectory Balance (conditional-$Z$) Objectives on BA-Shape Dataset.

|  | AUC | Accuracy |
| --- | --- | --- |
| Flow Matching | 0.99 | 0.99 |
| TB (uncon-$Z$) | 0.5~0.6 | - |
| TB (con-$Z$) | 0.5~0.8 | - |

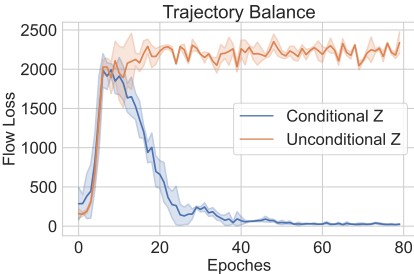 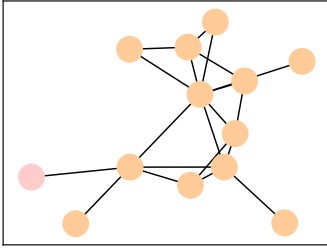

Figure 10: GFlowNets using trajectory balance. Left figure shows the different flow loss with conditional $Z_\theta(v_0)$ or unconditional $Z_\theta$. Right figure shows the explanation subgraph for one simple node with TB, which fails to find the motif. Without correct approximation to $Z$, GFlowExplainer could not sample $s_f$ so that $\mathcal{P}(s_f) \propto r(s_f)$.

we could not obtain correct $\mathcal{P}_F(-|-,-;\theta)$. In the graph structure data, most nodes have the same features (especially to the synthetic dataset), thus such conditional information does not distinguish them when feeding them into neural networks. For example, if $v_i$ and $v_j$ have the same features, we could output same $Z_\theta(v_i) = Z_\theta(v_j)$, while with high probabilities that $\sum r(s_f|v_i) \neq \sum r(s_f|v_j)$. Thus we have bias on approximations to the flow, which could further affect the approximations to $\mathcal{P}_F(-|-,-;\theta)$. However, using Flow matching loss, we have the following equation

$$\log \mathcal{L}(v_0) = \log \left( \frac{\sum_{T(s_t,a_t)=s_{t+1}} F(s_t, a_t|v_0)}{r(s_f|v_0) + \sum_{a_{t+1} \in \mathcal{A}} F(s_{t+1}, a_{t+1}|v_0)} \right). \tag{20}$$

For future work, the neighbor nodes of each $v_0$ could also pass message to it, thus aggregating them with more complex graph neural networks instead of MLP is more suitable in this study.

## F.2 QUALITATIVE ANALYSIS

### F.2.1 GRAPH-SST2 DATASET

We add a real-data set Graph-SST2 Yuan et al. (2022), which is a sentiment graph dataset for graph classification. It contains 70042 graphs with average 10 nodes in each graph. Each graph is labeled by its sentiment, which is either positive or negative. The node embeddings are initializes as the pre-trained BERT word embeddings. We train a GCN classifier with overall accuracy 88.7%.

Since the graph sizes are different and some of them just contain a few words, we choose graphs with relatively larger size to evaluate our explainability. We should note that this dataset does not have ground-truth structures, thus we visualize the subgraphs generated by GFlowExplainer for qualitative analysis. In Figure 11, each $s_f$ consists of green nodes, which are identified as important nodes for classification, the orange nodes are irrelevant nodes. Both graphs are correctly classified as "negative". We could find ("be failure", "because doesn't know to have fun") and ("it's frustrating to see these guys", "waste their talents ") could explain these classification decisions.

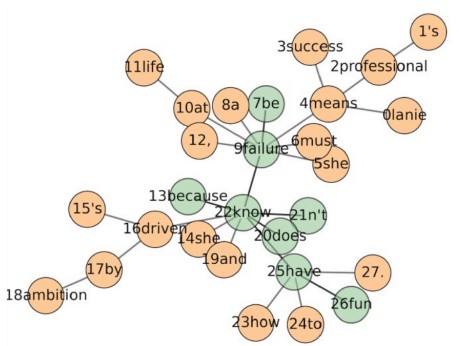
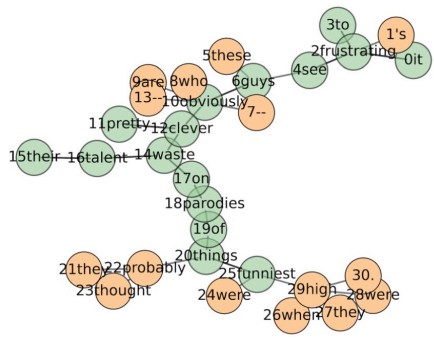

lanie's professional success means she must be a failure at life, because she's driven by ambition and doesn't know how to have fun.

it's frustrating to see these guys -- who are obviously pretty clever -- waste their talent on parodies of things they probably thought were funniest when they were high.

Figure 11: The subgraphs on the Graph-SST2 Dataset. The green nodes are identified as important nodes and the orange nodes are identified as irrelevant nodes.

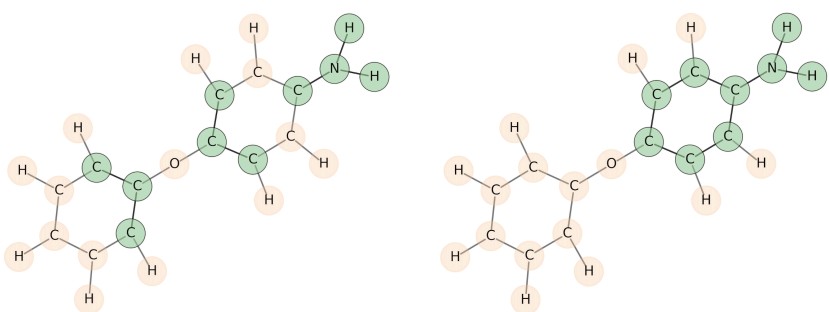

Figure 12: Qualitative Comparison between PGExplainer and GFlowExplainer on MUTAG dataset.

### F.2.2 MUTAG DATASET

We also visualize the results on MUTAG datasets in Figure 12, to show subgraphs are more intuitive and human-intelligible. It is known that the carbon rings and $NO_2$ or $NH_2$ groups are tend to be mutagenic. Our GFlowExplainer could identify these connected important components with correct classification. In contrast, the PGExplainer identifies discrete edges. In addition, GNNs utilize the message passing scheme to incorporate graph structures with node features. Our GFlowExplainer could construct the connected graphs by adding nodes from boundary of the current subgraph step-by-step, which is consistent with message passing scheme and provides more clear explanations.

### F.3 QUANTITATIVE COMPARISON

In this section, we compare GFlowExplainer with a shapley-value based approache SubgraphX Yuan et al. (2021) and DEGREE on accuracy. And also shows the fidelity and sparsity in our algorithm. The degree of fidelity assesses how closely related the explanations are to the model's predictions. It computes the difference between predictions with and without important structures. Sparsity measures the fraction of structures that are identified as important by explanation methods. Note that high Sparsity scores mean smaller structures are identified as important, which can affect the Fidelity scores since smaller structures (high Sparsity) tend to be less important (low Fidelity).

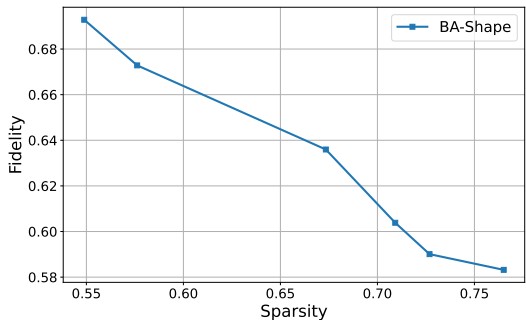

Figure 13: Sparsity and Fidelity in BA-Shape dataset

Table 7: Accuracy comparison

|  | BA-Shape | BA-Community |
|---|---|---|
| SubgraphX | 0.99 | 0.93 |
| DEGREE | 0.94 | 0.95 |
| Ours | 0.99 | 0.94 |

$$\text{sparsity} = \frac{1}{N}\sum_{i=1}^{N}(1 - \frac{|M_i|}{|G_i^L|}) \qquad (21)$$

$|M_i|$ denotes the number of important input features (nodes/edges/node features) identified. $|G_i^L|$ means the total number of features in $G_i^L$, which refers to the $L$-hop graph. For GFlowExplainer, the masks can be directly determined by the obtained subgraphs.

$$\text{fidelity} = \frac{1}{N}\sum_{i=1}^{N}(f(|G_i^L|)_{y_i} - f(|\hat{G}_i^L|)_{g_i}) \qquad (22)$$

Suppose $k$ is the number of edges(nodes) inside motifs for synthetic datasets, we will show the top$-k$ edges(nodes) ranked by their importance weights in our graph generation process. Based on the generation order of edges(nodes), we could assign different weights to them. In our GFlowExplainer, the weights of nodes have corresponding relationships with the their orders, which means the edges(nodes) with larger weights will be more likely to be added to the subgraph first.

As for the accuracy calculation, we follow similar setting in SubgraphX and DEGREE for fair comparison. We choose first $k$ nodes in each generated subgraph and check whether they are in the motif base and show the results in Table 7. In our implementations, we found the ground-truth indexes have some inconsistencies in each github public repository, making accuracy calculation biased, thus we fix the these inconsistencies by ourselves.

