# OpenReview forum: "DAG Matters! GFlowNets Enhanced Explainer for Graph Neural Networks"
_ICLR.cc/2023/Conference — ICLR 2023 poster_

### Official Review · Reviewer_NBTZ · 2022-10-17

[review text omitted: it was posted to a different submission]

---

> ### Author Response · Authors · 2022-11-19
> **Response to Reviewer NBTZ**
>
> Many thanks for your comments and we very appreciate your support for our paper.
>
> We have revised our related descriptions on conditional GFN.

---

### Official Review · Reviewer_CzDM · 2022-10-23

**Confidence:** 3
**Correctness:** 3
**Technical Novelty And Significance:** 3
**Empirical Novelty And Significance:** 2
**Recommendation:** 6

**Clarity, Quality, Novelty And Reproducibility:**

The paper is easy to follow, although the readers may need some background knowledge of GFlowNet to have a better understanding. The proposed algorithm provides an efficient method to find the parent state, which is easy to reimplement and can be easily extended to other scenarios. For the novelty, I'm a little concerned since the GFlowNet is borrowed from others' work. For the proposed algorithm, I'm also not sure whether it's novel enough.

**Strength And Weaknesses:**

The authors proposed a new framework based on GFlowNet. This network is more explainable since it's based on the conditional state flow network. The whole network is trained to minimize the discrepancy between the sum of the outgoing flow and the final reward and incoming flow. This framework is intuitive and novel. The authors also proposed an efficient algorithm to speed up the parent state exploration process.

For the weaknesses, I have some concerns regarding the experiment part and the design of the loss function.

Just like the authors claimed, the reward doesn't contain the sparsity or consistency part, which can be important in the GNN explainer model.

Secondly, I'm a little confused since the performance shown in the paper is somehow different from the ones in the PGExplainer. Like the BA-community/Tree-cycles/BA-2motifs, there is a huge gap. Could the authors explain why? I found the performance from the arxiv version here. https://arxiv.org/pdf/2011.04573v1.pdf . Meanwhile, they are some other state-of-the-art baselines that are not compared in the paper, like IB-SUBGRAPH, GraphMask, DIR(Discovering invariant rationales for graph neural networks).

I also have a question regarding the GFlowNet, since it is somehow related to reinforcement learning, I wonder if the model is robust. Or do we need to carefully tune the model to make it work?




**Summary Of The Paper:**

This paper proposed a method to explain the GNN using a newly proposed tool graph flow network(GFlowNets). This framework enables the model to connect the trajectories of the same graph even with different node sequences. The paper also proposed an efficient cut vertex criteria to speed up the whole process. Compared with traditional Tarjan methods, the proposed algorithm is much more efficient. Also, the experiment part shows that the result outperforms some other baselines.

**Summary Of The Review:**

This paper provides a new perspective for graph explainers. The paper also proposed an easy but efficient algorithm to find the parent state. My main concern is the gap in performance. Another concern is the novelty.

---

> ### Author Response · Authors · 2022-11-19
> **Response to Reviewer CzDM**
>
> Dear Reviewer CzDM,
>
> Thank your for your comments. We will make the modifications to the paper for a better understanding of our novelties based on your suggestions.
>
> $\textbf{Q1. Novelty of proposed algorithm / Is GFlowNets robust?}$
>
> 1. GFlowNet is a general framework (just like Reinforcement Learning) introduced at NeurIPS 2021. There are some applications using GFlowNets, such as molecule generation, causal discovery and discrete probabilistic modeling. Our paper is the first work to apply GFlowNets in the graph neural networks.
> 2. In addition, our paper is also the first work to apply state-conditional GFlowNets (the conditional concept was proposed only a few months ago in formal).
> 3. By taking advantages of flow matching loss objective in GFlowNets, which need explore all parent states for each current state, our work considers the subgraph as unordered set and constructs the flow structure as Direct Acyclic Graph instead of Tree structure. In the previous research(graph generation-based) there is no work like this. It is worthy to note that the DAG structure is not compulsory in GFlowNets training framework. We also conducted ablation study in Section 4.4 in our paper, to show tree structure using GFlowNets (GFlow-Sequence) performs much worse than our DAG structure method (GFlowExplainer).
> 4. Since both flow matching loss or trajectory balance loss(if uniform $\mathcal{P}_B$) in GFlowNets need to explore all parents, which is inefficient using the current method for exploring cut vertices, we proposed a new method to explore them in dynamic graphs, thus making our algorithm more efficient.
>
> In the previous related research, the Monte-Carlo tree search has high variance. GFlowExplainer aims to learn a policy so that $\mathcal{P}(s_f) \propto r(s_f)$. This is different to the previous reinforment learning aiming to maximize the reward, our algorithm could find diverse ``good enough'' candidates, thus preventing us from getting stuck in local optima. In our inductive experiments in Section 4.4, our performance is much better than others, which could show our algorithm is more robust.
>
> $\textbf{Q2. Need carefully tune the model to make it work}$
>
> The neural network models in our paper are mainly for explanations for graph neural networks. First is the APPNP for learning the feature representation,  the message passing scheme when generating the subgraph by adding boundary nodes. The action function and stopping criteria are flexible in GFlowNets.  For GFlowNets, the main part is the loss function (flow matching loss in our case). Therefore, we do not need very  carefully tune the model ( on GFlowNets)  to make it work. The choice of NNs depend on the specific task or the application.
>
> $\textbf{Q3. Baselines and Performance Gap? }$
>
> Please refer to the general response at https://openreview.net/forum?id=jgmuRzM-sb6&noteId=2MyVAMlEQHb.
>
> Hope we have addressed your concerns.

---

> > ### Author Response · Authors · 2022-11-26
> > **Look forward to your reply**
> >
> > We hope our response to the performance gap and novelty of using GFlowNets could address your concerns.
> >
> > We are the first work to apply the conditional GFlowNets into GNN fields, and we also provide efficient parent exploration to accelerate the training process and a more suitable policy network ( including both features representation in states and action ). We also conduct extensive experiments in both the paper and the appendix to show the robustness of the GFlowNets framework.
> >
> > Since the discussion window between reviewers and authors is ending, we would appreciate it if you could raise your score to boost our chance of more exposure to the community. Of course, we are also welcome to have further discussions to improve our paper if there is any other concern.
> >
> > We look forward to your reply.

---

> > > ### Comment · Reviewer_CzDM · 2022-11-26
> > > **Reply to the authors**
> > >
> > > After reading all of these replies, I think the authors solve my main concern regarding the performance gap, more extensive experiments, and other baseline models. I'm willing to raise my score.

---

> > > > ### Author Response · Authors · 2022-11-30
> > > > **Thank you**
> > > >
> > > > Dear Reviewer CzDM,
> > > >
> > > > Thank you for your time reading our response and for appreciating our revision. We are glad to address your main concerns and will keep polishing our manuscripts.

---

### Official Review · Reviewer_SPuf · 2022-10-24

**Confidence:** 2
**Correctness:** 3
**Technical Novelty And Significance:** 3
**Empirical Novelty And Significance:** 2
**Recommendation:** 6

**Clarity, Quality, Novelty And Reproducibility:**

Clarity: This paper does not provide a clear motivation for the proposed strategy; besides, how the proposed strategy avoids the disadvantages of previous works mentioned in the introduction should be further elaborated.

Quality: The adopted techniques are sound in this paper, but the organization of this paper can be further improved – see Summary of The Review for more details.

Novelty: This paper is novel in techniques under the topic of GNN interpretation.

Reproducibility: Most of the codes are provided. All datasets are open source.


**Strength And Weaknesses:**

Strength:

(1) This paper is solid in techniques: definitions are clearly presented, and proofs are also provided in the appendix.

(2) A novel approach is proposed borrowing the idea of generative flow networks to obtain subgraphs as interpretations for GNNs, which could reveal novel insights under this research topic.

(3) The empirical performance superiority on most datasets seems to be promising.

Weaknesses:

(1) The motivation of this paper is unclear: why do we need this model considering the existing GNN interpretation approaches? The presented motivations are not convincing – see Summary of The Review for more details.

(2) There are certain phenomena not clarified in the experiments – see Summary of The Review for more details.

(3) Only one real-world dataset is involved in the experiments performed in this paper.

(4) The writing of this paper needs to be further polished. I did notice some typos, such as “GFlowExplainer does not pre-training process”.


**Summary Of The Paper:**

This paper proposes a novel strategy to obtain a connected subgraph as the interpretation of node classification and graph classification tasks. The proposed strategy starts from a state with only one node and learns a policy to achieve state transition via sampling new nodes and adding them to the current state. In general, the techniques used in this paper are novel under the topic of GNN interpretation, which reveals further insights on how to understand in what ways GNN predictions are made. However, there are also clear disadvantages in this paper. See the Summary of The Review for more details.

**Summary Of The Review:**

This paper proposes a strategy inheriting the idea of Generative Flow Networks to generate subgraphs as interpretations for GNN predictions, which is novel in techniques. Meanwhile, the theorems and lemmas are supported with detailed proofs. However, there are multiple unaddressed questions that prevent this paper from being well-prepared for publication. For example, the author mentioned that the primary motivation of this paper is: (1) searching subgraphs is a combinatorial problem, and it is hard to solve; and (2) previous works model the selection of subgraphs as a sequence generation problem, which is not beneficial. However, first, most GNN interpretation works based on learning do not require solving any combinatorial problem (e.g., [1]); second, we may also regard the proposed approach in this paper based on state transitioning as a strategy based on sequence generation. I thus believe the motivations above deserve a more rigorous justification. Moreover, there is no detailed discussion on how to choose an appropriate stopping criterion. Should all generated subgraphs reach the constraint of size K_M? Will there be cases where the size of the underlying interpretation subgraph is different from each other? If they exist, how could we design an appropriate stopping criterion that fits all scenarios?

Finally, I would still have extra concerns as follows.

(1) There is a policy network in Fig. 1, while this notion is never mentioned elsewhere in this paper.

(2) The reported AUC seems to be much worse compared with the reported accuracy in the vanilla papers of baselines (e.g., [1]). Will it be more comprehensive to compare both AUC and accuracy?

(3) There is no evidence showing that the exhibited superiority is (partially) because the proposed strategy does not generate subgraphs in a sequential manner.

(4) Will it be better to introduce the definitions in Section 3.3 before these notions are referred to?

(5) Is there any specific reason for the unique performance inferiority on BA-Community datasets compared with the baselines?

(6) Is it possible to include more than one real-world dataset for more comprehensive experiments?

[1] Rex Ying et al. Gnnexplainer: Generating explanations for graph neural networks. NeurIPS 2019.

---

> ### Author Response · Authors · 2022-11-19
> **Response to Reviewer SPuf ( Part 1 )**
>
> Dear Reviewer SPuf,
>
> Thanks for the suggestions that helped us make the paper more rigorous, we will polish our draft based on your comments. Some questions and weaknesses revolve around one theme, so we'll put answers together.
>
> $\textbf{Q1.Motivation of this paper. Why GFlowNets and why graph generation?}$
>
> Some GNN interpretation works(post-hoc) do not need to solve combinatorial problem, such as GNNExplainer and PGExplainer. However, they cannot guarantee that nodes and edges in the outputs are connected, thus the explanatory subgraphs cannot visualize the message passing paths in GNNs. Therefore, there is another research branch considering generating connected subgraphs for interpretations, which are more intuitive and human-intelligible, such as SubgraphX, Causal Screening and RGExplainer. Therefore, for each node or graph to be interpreted, the objective is to solve the combinatorial problem over graphs. Our work is also based on graph generations, to construct connected explanatory subgraphs.
>
> SubgraphX and Causal Screening use search-based methods with designed criteria to solve the optimization problem. However, SubgraphX constructs the Monte-Carlo tree and uses Shapley Value to compute the scores, which is very time inefficient. Causal Screening tries to interpret from the causal perspective while there is not a rigorous explanation of ''confounding''. As for RGExplainer, which takes advantage of the strong search abilities of Reinforcement Learning, constructs the graph generation as a sequence modeling. However, it needs a pre-training process for graph generation to cover all possible generated orderings for an explanatory graph, which is not efficient.
>
> Therefore, GFlowExplainer aims to make progress in GNN explanation from the graph generation perspective, and it could indeed overcome the current predicaments mentioned above. It has strong exploration ability and could prevent us from getting stuck in local optima. Different from Reinforcement Learning approach, it is more robust ( we show the experiments in the inductive setting in Section 4.4 in our paper to convince this statement).We construct the flow structure as Direct Acyclic Graph, which could also eliminate the pre-training process. However, without flow matching loss like in GFlowNets, RL method could not learn such a generator policy so that $P(s_f)\propto r(s_f)$ in the DAG structure. Compared to SubgraphX, we do not use the high computational Shapley value.
>
> $\textbf{Q2. Proposed strategy does not generate subgraphs in a sequential manner?}$
>
> We will give a more rigorous explanation of the sequential manner for  our task: GFlowExplainer takes a state transition by adding a node in the current subgraph. The compositional subgraph $s_f$ is thus adding by sequentially. However, in this task, the graph actually should be $\textbf{unordered}$. For a connected graph, unless some nodes are ``connected" due to the previous action, the order of adding nodes does not affect its construction. The sequence modeling in RGExplainer and SubgraphX follows the tree structure, while we consider this as a directed acyclic graph structure (at least one trajectory attaining one state ). Therefore, we want to differentiate our approach from traditional sequence modeling approaches, since ours eliminate the effects of the sequence; our state is an unordered set instead of an ordered sequence ( or topology )
>
> We did the ablation study in Section 4.4. GFlow-Sequence has the same setting as GFlowExplainer except for the structures. GFlow-Sequence generates the connected graph sequentially, and each state represents an ordered sequence; therefore, for each state $s_t, s_t\neq s_0$, there is only one parent state.
> For example, suppose a connected graph $s_t$ is [400,401,402]; its parent state $s_{t-1}$ could only be [400,401] due to the sequential manner. In contrast, with GFlowExplainer, its parent states are [400,401] and [400,402] (If there is no cut vertex). In Figure 4 in Section 4.4 and Figure 9 in Appendix D.4,  we can find in GFlowExplainer has better AUC than GFlow-Sequence, which could convince the statement that the superiority is partial because the proposed strategy considers the unordered property of the graph.
>
> $\textbf{Q3. What is policy nework?}$
>
> Sorry we made a mistake on its definition. It refers to the neural networks we want to train (parameters in Eq(8)$\sim$Eq(13)) in the experiments. And in the paper it refers to $P_F(s_{t+1}|s_t)$. We have updated paper for this.

---

> > ### Author Response · Authors · 2022-11-19
> > **Response to Reviewer SPuf ( Part 2 )**
> >
> > $\textbf{Q4. Explaination on the stopping criteria}$
> >
> > We impose a constraint $|s_f| \leq K_M$, thereby $s_f$ has at most $K_M$ nodes. In our paper, we also introduce a self-attention mechanism, i.e., Eq(12) and Eq(13), so that when our policy network samples ``EOS''( which is additional action in our task) , it meets the stopping criteria and attains the terminal states. Therefore, in our experiments, not all generated subgraphs reach the constraint of size $K_M$.
> >
> > We can also introduce some regularization terms in the reward function Eq(10) to restrict the characteristics of the explanatory subgraph, to better train the parameters in Eq(12) and Eq(13). For example, adding penalties $c \times |s_f|$ on the graph size $|s_f|$, in which $c$ is a hyper-parameter and it depends on different graphs. In our experiments, this mechanism could help to fit different scenarios.
> >
> > It is worthy to say we are not the first work to use $K_M$. For example, GNNExplainer imposes a constraint on graph`s size $|G_S| \leq K_M$ so that the subgraph has most $K_M$ nodes. For the SubgraphX, it uses the same number to control the maximum number of nodes in the explanations for all methods. For DEGREE, it scores each component and chooses the $k-$top highest score nodes for explanations(they choose $k$ as the size of the motif base). However, in real-world sometimes we do not know the ground-truths.
> >
> >
> > $\textbf{Q5. Why BA-Community datasets performed worse than others ? }$
> >
> > We guess the reason behind is due to the generated graph size since each trajectory could sample different subgraphs with the stopping criteria we defined in our paper) .  In BA-Community, the ground-truth has different size for each node (even though the motif base is still the house structure). However, for BA-Shape, Tree-Cycles and Tree-Grid datasets, they have the same size of ground-truth nodes respectively, which is relatively easier than BA-Community when generating subgraphs sequentially.
> > DEGREE directly decomposes the influence of node groups in the forward pass, which is not an approximation-based or perturbation-based approach. DEGREE attains SOTA in the decomposition-based approach, while the computational complexity is high (about 2s to compute the subgraph for a node) Therefore, even though we have slightly worse performance on AUC, our time efficiency is better than DEGREE. In addition, since it fixes the graph size (same as the motif base) while in graph generation we do not have this constraint ( we just set the maximum size with 20), we are more likely to have a larger size of the subgraph.

---

> > > ### Comment · Reviewer_SPuf · 2022-11-24
> > > **Response to author(s) from reviewer SPuf**
> > >
> > > Thanks for the detailed reply. I believe this reply addresses most of my doubts, but I still have a follow-up concern: in the answer “Q5”, what do you mean by “the ground-truth has different sizes for each node (even though the motif base is still the house structure)”? In my mind, the role of the node can not be determined until the interpretation method identifies the house shape accurately. Even if identifying triangles/rectangles already provides the clue to the role of a specific node, such a role is still not valid until the whole house shape is identified. Then why ground truth has different sizes in this case?

---

> > > > ### Author Response · Authors · 2022-11-24
> > > > **Response to reviewer SPuf**
> > > >
> > > > Dear Reviewer SPuf,
> > > >
> > > > Thanks for your reply, and we are glad to know most of your concerns have been addressed.
> > > >
> > > > Yes, we want to identify the motif structure for this synthetic dataset. But we find there is a slight inconsistency when loading the pkl file dataset; we use the edge_label_matrix attribute to develop the ground-truth based on previous python implementation, which is slightly different from DEGREE. Sorry for this experiment inconsistency; we have fixed it, our accuracy has improved if we follow the same setting in DEGREE, and we will add it to our final manuscripts. We also checked all other experiments and the remaining are consistent.
> > > >
> > > > As for the AUC calculation, even though the house shape is identified, when calculating AUC, our graph-generation-based approach calculates the score based on the weights ( the node added to the subgraph set first will have larger weight ). Therefore, even if some nodes are not in the motif structure, they are also assigned weights based on their ordering to be added, thereby decreasing the AUC score. Therefore, we show the qualitative analysis for our graph to show GFlowExplainer could still identify the house shape accurately. You can find the qualitative analysis (Figure 2) in the BA-Community dataset; both approaches will find more irrelevant nodes (orange) than other node classification tasks, thus making the AUC calculation worse than other datasets.
> > > >
> > > > As for the DEGREE algorithm, we checked with the code; it calculates the accuracy by sorting the score of each node and choosing top-5 to see whether each one is in the motif base. And for AUC calculation, since it is decomposed approach,  its node score distribution is relatively more extreme than ours ( the irrelevant nodes have almost a value of 0 ), thus the AUC is better for ours in this case.
> > > >
> > > > For graph generation, the smart stopping criteria are important to prevent generating very large subgraphs, potentially controlling the dilemma when adding irrelevant nodes. We use the attention mechanism to handle this problem and as we mentioned for Q4, a larger penalty could also help with this. But as we clarify, both decompose-based and graph-generation based are different research branches; for efficiency and overall accuracy, GFlowExplainer performs better.
> > > >
> > > > We hope our answers address your concern. And we are still active to looking forward to your reply.

---

> > > > > ### Comment · Reviewer_SPuf · 2022-11-25
> > > > > **Response to author(s) from reviewer SPuf**
> > > > >
> > > > > Thanks for the detailed reply. I have improved and finalized my score.

---

> > > > > > ### Author Response · Authors · 2022-11-25
> > > > > > **Thank you**
> > > > > >
> > > > > > We sincerely thank the reviewer for raising the score. Your constructive comments indeed help improve our paper.

---

### Official Review · Reviewer_uNyQ · 2022-10-24

**Confidence:** 3
**Correctness:** 3
**Technical Novelty And Significance:** 2
**Empirical Novelty And Significance:** 3
**Recommendation:** 6

**Clarity, Quality, Novelty And Reproducibility:**

The paper is well written. The paper is not novel from a theoretical perspective but is a use case of GFlowNet in a new domain, i.e. explainability for GNNs.

As I'm not that familiar with the explainability of GNNs, it is hard for me to say that much regarding related works and experiments. But I felt  the paper missed a few recent works in this domain and evaluation metrics are not complete.

**Strength And Weaknesses:**

**Pros:**

- Applying GFlowNet in a new domain, i.e. GNNs.

**Cons:**

- Based on the loss in eq. 18, the authors only use the naive "flow matching condition" which is underspecified and tends to have biased to smaller trajectories. Besides that, it is computationally very inefficient. These are already known and new methods have been proposed.

- This also raises a question about the quantitative results. To me, it is not clear if the shown results in Figure 3 which show a smaller sub-graph for the proposed method are due to these biases or if it is really beneficial from GFlowNet. I would rather like to see the results for the "trajectory balance" based loss.

- I'm also not sure how much gain they could get from the efficient parent state explorations. An ablation study around it would be gratefully appreciated. I think a naive solution still should not be that complicated.

- Evaluation metrics are not enough. I would like to see comparisons in terms of Fidelity and Sparsity as well.

- I also believe some of the baselines are missed.



**Summary Of The Paper:**

The authors propose to use GFlowNet in a new application, explainability for GNNs. To me, this is a resemble application for GFlowNet, though, I'm not that much familiar with the explainability problem of GNNs. Intuitively, GFLowNet is able to generate subgraphs based on learning multiple trajectories which maximize a specific reward. In this paper, the authors use MI as the reward which seems to be consistent with the previous models in this domain.

The paper is fairly well-structured and easy to follow. The most of methodological part of the paper is repeating the first GFLowNet paper. My main concern with this paper is that the authors use "flow matching condition" that has known issues. So, it raised some questions regarding some of the qualitative results.



**Summary Of The Review:**

The paper seems to be a good application paper, however, it needs some modification in terms of the loss as well as experiments.

---

> ### Author Response · Authors · 2022-11-19
> **Response to Reviewer uNyQ**
>
> Dear Reviewer uNyQ,
>
> Thanks for your comments. Hope the following answers and our additional experiments could address your concerns.
>
> $\textbf{Q1. Trajectory Balance Loss}$
>
> Thanks for pointing out this new objective function. We have checked with the paper and the official code. We note that for Trajectory Balance objective, the natural choice for the backward policy is to set $\mathcal{P}_B(\cdot|s_t)$ to be uniform over all the parents of a state $s_t$, i.e., $P_B(\cdot |s) = 1/\\#[ {s | (s\ \rightarrow s_t)\in \mathcal{A}\}]$. This means we do not need calculate the inflow from all parents based on flow matching condition, which could be more efficient and effectiveness. But the parent explorations still matter in this work, we need to find the cut vertices for each state, and $\mathcal{P}_B(\cdot |s_t) = 1/(|s_t| - m-1)$, where $m$ is the number of cut vertices. Additional minus 1 is due to the node to be interpreted could not be deleted as well.
> But the trajectory balance loss may not work in conditional GFN, or it is not trivial to train a good policy network. In trajectory balance, we need to parameterize the $\hat{Z}_\theta$, which represents the total flow in the DAG flow structure.
>
> In the state-conditional case,  we should approximate the $\textbf{total flow for subgraphs}$ starting from our conditional information, which makes TB-based approach more difficult to approximate $\mathcal{P}_F$.  In paper [1], the authors propose the flow matching loss for conditional cases, and [2] did not consider conditional TB. We conducted both unconditional $\hat{Z}_\theta$ and conditional $\hat{Z}_\theta(\cdot)$ in our updated draft. You could find the discussions in Appendix F.  In our experiments, we find the flow matching loss condition performs the best and also does not sacrifice much time.
>
> As for the results which show smaller subgraphs, it is because we introduce the specific stopping criteria, and based on the learning process, we could find the motif structure on our policy. In our experiments, we find the motif structure is always identified first in our graph generation.
>
> [1] GFlowNets Foundations
>
> [2] Trajectory balance: improved credit assignment in gflownets
>
> $\textbf{Q2. Benefits from efficient parent state explorations}$
>
> The insight of our parent exploration method comes from the experiments on MUTAG dataset, when we used Tarjan's approach, there raised some system errors since each graph size is relatively larger than other datasets. With larger size graph, it is very inefficient to use previous method.
>
> We did the comparisons on efficiency between our theorem and Tarjan's approach in Appendix D.3 with small graph size. It shows that with the increasing size of the graph, the accumulated time of Tarjan’s algorithm increases sharply than ours.
> We will conduct an ablation study on Tarjan's approach, our GFlowExplainer(FM) and GFlowExplainer (TB) in the experiments and report the time in the table in our final draft.

---

> > ### Comment · Reviewer_uNyQ · 2022-11-26
> > **Thanks for the response**
> >
> > The authors have addressed my main concern and I have updated my score based on that.

---

> > > ### Author Response · Authors · 2022-11-30
> > > **Thank you**
> > >
> > > Dear Reviewer uNyQ,
> > >
> > > Thank you for your time reading our response and for appreciating our revision.

---

### Author Response · Authors · 2022-11-19
**General Response**

Dear reviewers, thanks for your constructive comments and suggestions for our paper. We will respond to all your questions and hope our answers can address your concerns. Since there are some overlap questions, we will answer them in the general response.

$\textbf{There are some updates in our paper (highlighted in blue)}:$
1. We add trajectory balance with both conditional $Z_\theta(v_0)$ and unconditional $Z_\theta$ to train our GFlowExplainer (please refer to Appendix F.1).
2. We add a real-world dataset: Graph-SST2, for graph classification to show the explanation abilities of GFlowExplainer from the qualitative aspect. We also visualize the qualitative analysis on MUTAG datasets to show the difference between graph generation-based approach and the perturbation-based approach (please refer to Appendix F.2 and Appendix F.3).
3. We add sparsity and accuracy as additional metrics. Since some baselines have high computational cost, we are not be able to complete all the experiments within this window period. We will keep adding experiments to our manuscripts (please refer to Appendix F.4).
4. There are some changes in the introduction and problem formulation to clearly show our motivations.


$\textbf{Q1. Missing some baselines ?}:$

To the best of our knowledge, our paper has covered the most representative comparable approaches. Some mentioned approaches cannot be used for comparison in our paper.

There are different research lines of rationalization. One is the post-hoc explainability attributing a model's prediction to the input graph with a separate explanation method. The other is intrinsic interpretability, which incorporates a rationalization module into the model to make transparent predictions. DIR-GNN [1] is an intrinsic interpretation method. IB-SUBGRAPH [2] mainly aims to improve the model predictions, and its interpretation is out of scope in our task. GraphMask [3] uses a similar idea to PGExplainer, but it studies the edges in every GNN layer, while PGExplainer only focuses on the input space. Therefore, they could not be directly applied to comparisons in our work (You can also find that some of these papers do not have the same baselines as in ours). Here we will add SubgraphX [4] as an additional baseline because it is a post-hoc interpretation method and as well as a graph generation-based. GNN-LRP [5] is a decomposed-based approach, and we have compared a more representative and better performance approach DEGREE [6] in our work.

[1] Discovering Invariant Rationales for Graph Neural Networks

[2] Graph Information Bottleneck for Subgraph Recognition

[3] Interpreting Graph Neural Networks for NLP With Differentiable Edge Masking

[4] On Explainability of Graph Neural Networks via Subgraph Explorations

[5] Higher-Order Explanations of Graph Neural Networks via Relevant Walks

[6] Decomposition based explanation for graph neural networks

$\textbf{Q2. Why there is a large performance gap?}:$

The official code[1] is Tensorflow version and has some implementation issues in the original codebase. Thus we followed the code on the pytorch implementations[2] linked in the [1]'s Github (https://github.com/flyingdoog/PGExplainer). Since the authors endorsed these implementations, and the reproduction had been formally peer-reviewed, and discussed with the original authors, we believed that this pytorch version is a high-quality code implementation and a fair benchmark for comparison, thus just applying it without further modifications.

For fairness comparison, we use the same framework to apply GNNExplainer, PGExplainer and RGExplainer. Our GFlowExplainer is also built on their implementations. The GNN/GCN models, datasets, and evaluation metrics are the same. Thus the comparisons are fair.

[1] Parameterized Explainer for Graph Neural Network( https://arxiv.org/abs/2011.04573)

[2] https://github.com/LarsHoldijk/RE-ParameterizedExplainerForGraphNeuralNetworks

---

### Author Response · Authors · 2023-10-08
**Code implementation**

Hi all, you can find the official code implementation on https://github.com/muz1lee/gflowexplainer/tree/main

---

### Decision · Program_Chairs · 2023-01-20

**Decision:**

Accept: poster

**Justification For Why Not Higher Score:**

While the paper ticks all the boxes for acceptance, I found that, as with any such emerging proposal, there are questions raised over whether the generative approach is really the best way towards achieving GNN explainability. This is reflected in the reviewers' initial comments and, while the authors certainly handled the rebuttal adequately to secure acceptance, I find that this line of research would need more iterations of fine-tuning before it would be spotlight-worthy. I hope the advice and ponderings provided by the reviewers, as well as the comments the work will receive at ICLR, can aid the authors on this quest going forward!

**Justification For Why Not Lower Score:**

There is a clear consensus between the reviewers, and the work is interesting to a broad ICLR audience.

**Metareview: Summary, Strengths And Weaknesses:**

This work proposes to use GFlowNets to propel GNN explainability. Both topics are very interesting and timely, and after a round of rebuttal discussions, all reviewers agree that the paper is worth accepting. I fully agree with them, and support this paper for acceptance.

**Note From Pc:**

if the above contains the word "oral" or "spotlight" please see: "oral" presentation means -> notable-top-5% and "spotlight" means -> notable-top-25%. As stated in our emails, we are disassociating presentation type from AC recommendations